Perspective

# Emerging opportunities and challenges for the future of reservoir computing

Min Yan ®[1], Can Huang ®[1] ✉, Peter Bienstman[2], Peter Tino[3], Wei Lin ®[4,5] &
Jie Sun ®[1] ✉

Reservoir computing originates in the early 2000s, the core idea being to utilize dynamical systems as reservoirs (nonlinear generalizations of standard bases) to adaptively learn spatiotemporal features and hidden patterns in complex time series. Shown to have the potential of achieving higher-precision prediction in chaotic systems, those pioneering works led to a great amount of interest and follow-ups in the community of nonlinear dynamics and complex systems. To unlock the full capabilities of reservoir computing towards a fast, lightweight, and significantly more interpretable learning framework for temporal dynamical systems, substantially more research is needed. This Perspective intends to elucidate the parallel progress of mathematical theory, algorithm design and experimental realizations of reservoir computing, and identify emerging opportunities as well as existing challenges for large-scale industrial adoption of reservoir computing, together with a few ideas and viewpoints on how some of those challenges might be resolved with joint efforts by academic and industrial researchers across multiple disciplines.

At the core of today's technological challenges is the ability to process information at massively superior speed and accuracy. Despite large-scale success of deep learning approaches in producing exciting new possibilities[1–7], such methods generally rely on training big models of neural networks posing severe limitations on their deployment in the most common applications[8]. In fact, there is a growing demand for developing small, lightweight models that are capable of fast inference and also fast adaptation - inspired by the fact that biological systems such as human brains are able to accomplish highly accurate and reliable information processing across different scenarios while costing only a tiny fraction of the energy that would have been needed using big neural networks.

As an alternative direction to the current deep learning paradigm, research into the so-called neuromorphic computing has been attracting significant interest[9]. Neuromorphic computing generally focuses on developing novel types of computing systems that operate at a fraction of the energy comparing against current transistor-based computers, often deviating from the von-Neumann architecture and drawing inspirations from biological and physical principles[10]. Within the broader field of neuromorphic computing, an important family of models known as reservoir computing (RC) has progressed significantly over the past two decades[11,12]. RC conceptualizes how a brain-like system operates, with a core three-layer architecture (see Box 1 and Box 2): An input (sensing) layer which receives information and performs some pre-processing, a middle (processing) layer typically defined by some nonlinear recurrent network dynamics with input signals acting as stimulus and an output (control) layer that recombines signals from the processing layer to produce the final output. Reminiscent of many biological neuronal systems, the front end of an RC network, including its input and processing layers, is fixed and non-adaptive, which transforms input signals before reaching the output layer; in the last, output part of an RC the signals are combined in some optimized way to achieve the desired task. An important aspect of the output layer is its simplicity, where typically a weighted sum is

[1]Theory Lab, Central Research Institute, 2012 Labs, Huawei Technologies Co. Ltd., Hong Kong SAR, China. [2]Photonics Research Group, Department of Information Technology, Ghent University, Gent, Belgium. [3]School of Computer Science, The University of Birmingham, Birmingham B15 2TT, United Kingdom. [4]Research Institute of Intelligent Complex Systems, Fudan University, Shanghai 200433, China. [5]School of Mathematical Sciences, SCMS, SCAM, and CCSB, Fudan University, Shanghai 200433, China. ✉ e-mail: huangcan321@gmail.com; riosun@gmail.com

## BOX 1

# Comparison bettwen deep learning and reservoir computing

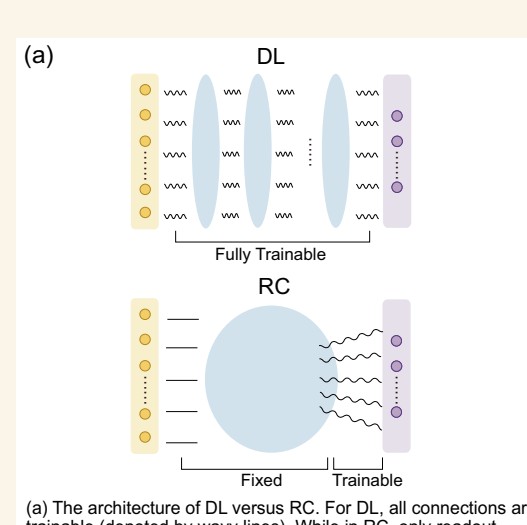

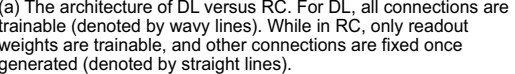

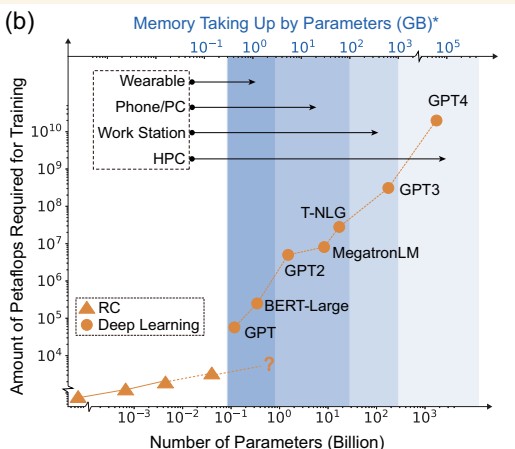

(a) The architecture of DL versus RC. For DL, all connections are trainable (denoted by wavy lines). While in RC, only readout weights are trainable, and other connections are fixed once generated (denoted by straight lines).

(b) RC versus DL in terms of the number of parameters and computational cost (measured in the amount of petaflops) required for training [128].
* Estimation of memory storage of the parameters, assuming that each parameter is in Float32 format.

Deep learning (DL) and reservoir computing (RC) are both machine learning techniques. They share some common characteristics. For example, both of them are data-driven frameworks for learning, taking inputs and transform them (nonlinearly) to match desired outputs. By learning the features from the input data, they are shown to be universal function approximation, so as to fulfill sophisticated tasks.

However, deep learning and reservoir compuitng are different in some degrees:
1. Architecture design: DL and RC can be distinguished directly from their structures. As shown in Fig. (a), in DL, all the parameters are fully trainable, namely all connections are continuously updated during the training phase. While in RC, only the readout weights are trained. Other connections among neurons are fixed once generated and are not updated any further. This structural difference indicates that RC usually has smaller parameter size than those of DL.

2. Training procedure: Different architectures determine that DL and RC are trained distinctly. In DL, there have been many training algorithms and tools developed, such as backpropagation (BP), stochastic gradient descent (SGD), Newton's method (NM), and so on. However, in RC, it is simple regression (e.g., linear regression, Lasso regression and ridge regression) that are usually adopted in training. The small parameter size and simple training procedure of RC together lead to much less training time and resource consumption.

3. Model complexity & performance: DL and RC have distinct model size, training comlexity and performance. In Fig. (b), we summrize the parameter size and required training petaflops of DL and RC. As the capacity of deep learning increases, the parameter size also grows, which is a challenge for practical application. For example, the memory of smart watch is around typically 2GB, so it can be equipped with GPT (~0.5 GB) or BERT-Large (~1.3 GB). For large networks such as GPT3 (~652 GB) and GPT4 (~6557 GB), only workstations or high performance cluster (HPC) can incorporate them. Inversely, since RC has much less parameters, it can be applied on diverse devices flexibly. Is has been shown that RC can realize image recognition with around $10^{-5}$ petaflops [129], indicating wide scope for further explorations. In addition, although parameter size is smaller, RC is utilized in improving the accuracy in climate modeling [110] and fulfilling weather forecast [111], which had been realized by deep learning previously.

As one of the most popular machine learning algorithms, DL has been studied widely. Nevertheless, RC seems to remain at primary stage, no matter in theoretical or algorithm level. It is still an open question where the full potential of RC is (as indicated by a question mark in Fig. (b)), and how is the training complexity if RC involves around billions of parameters, for which we draw our hypothesis in dotted lines in Fig (b).

## BOX 2
# Schematic representation of the reservoir computing (RC) framework

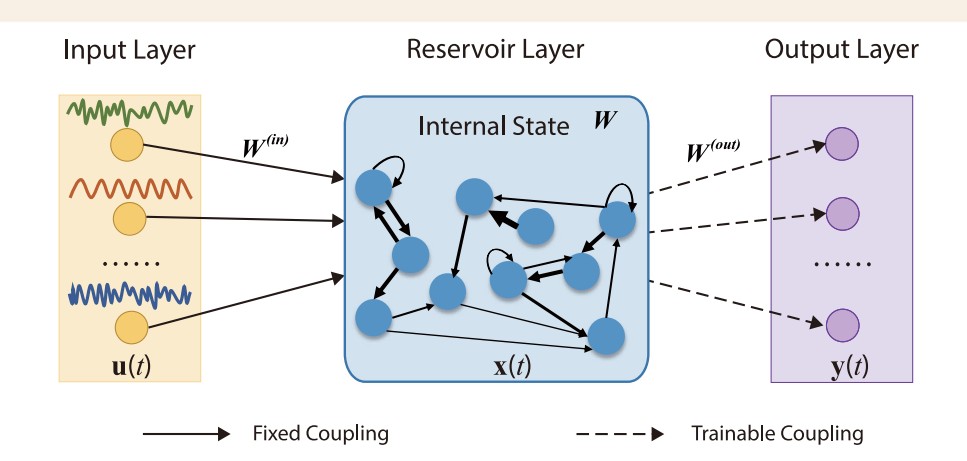

The input and output of an RC are typically temporal sequences (time series), denoted here as {**u**(t)} and {**y**(t)}, respectively. The raw inputs are transformed and coupled to the internal part of RC, which itself is a coupled dynamical system with multiple nodes forming a recurrent network. The internal states of the dynamical system are computed following pre-determined computational rules and at the output part re-combined into the RC outputs. In a typical setup, the input-to-internal as well as the internal parts of RC are all fixed and only the internal-to-ou tput part is trained with data-contrasting to general RNNs where the former parts are also trained.

sufficient, reminding a great deal of how common mechanical and electrical systems operate - with a complicated core that operates internally and a control layer that enables simple adaptation according to the specific application scenario.

Can such an architecture work? This inquiry was attempted in the early 2000s by Jaeger (*echo state networks (ESNs)*[11]) and Maass (*liquid state machines (LSMs)*,[12]), achieving surprisingly high level of prediction accuracy in systems that exhibit strong nonlinearity and chaotic behavior. These two initially distinct lines of work were later reconciled into a unified, reservoir computing framework by Schrauwen and Verstraeten[13], explicitly defining a new area of research that touches upon nonlinear dynamics, complex networks and machine learning. Research in RC over the past twenty years has produced significant results in the mathematical theory, computational methods as well as experimental prototypes and realizations, summarized in Fig. 1. Despite successes in those respective directions, large-scale industry-wide adoption of RC or broadly convincing "killer-applications" beyond synthetic and lab experiments are still not available. This is not due to the lack of potential applications. In fact, thanks to its compact design and fast training, RC has long been sought as an ideal solution in many industry-level signal processing and learning tasks including nonlinear distortion compensation in optical communications, real-time speech recognition, active noise control, among others. For practical applications, an integrated RC approach is much needed and can hardly be derived from existing work that focuses on either the algorithm or the experiment alone. This perspective offers a unified

overview of the current status in theoretical, algorithmic and experimental RCs, to identify critical gaps that prevents industry adoption of RC and to discuss remedies.

## Theory and algorithm design of RC systems
The core idea of RC is to design and use a dynamical system as reservoir that adaptively generates signal basis according to the input data and combines them in some optimal way to mimic the dynamic behavior of a desired process. Under this angle, we review and discuss important results on representing, designing and analyzing RC systems.

### Mathematical representation of an RC system
The mathematical abstraction of an RC can generally be described in the language of dynamical systems, as follows. Consider a coupled system of equations

$$\begin{cases} \Delta x = F(x; u; p), \\ y = G(x; u; q). \end{cases} \quad (1)$$

Here the operator $\Delta$ acting on $x$ becomes $\frac{dx}{dt}$ for a continuous-time system, $x(t+1) - x(t)$ for a discrete-time system, and a compound of these two operations for a hybrid system. Additionally, $u \in \mathbb{R}^d$, $x \in \mathbb{R}^n$, and $y \in \mathbb{R}^m$ are generally referred to as the input, internal state and output of the system, respectively, with vector field $F$, output function $G$ and parameters $p$ (fixed) and $q$ (learnable) representing

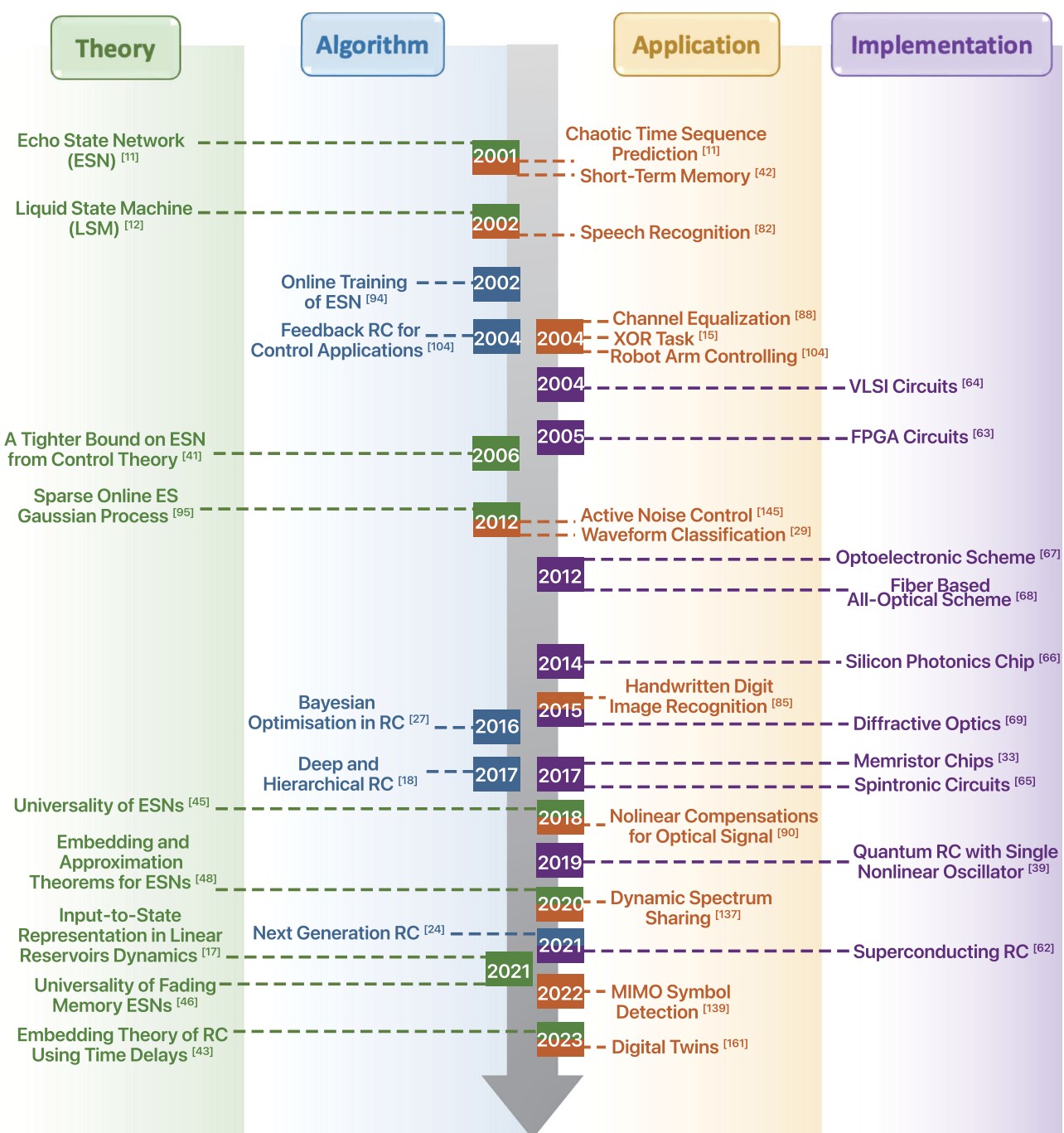

**Fig. 1 | Selected research milestones of RC encompassing system and algorithm designs, representing theory, experimental realizations as well as applications.** For each category a selection of the representative publications were highlighted.

their functional couplings. Once set up by fixing the vector field $F$ and the output function $G$ and the parameters $p$, one can utilize the RC system to perform learning tasks, typically in time-series data. Given a time series $\{z(t) \in \mathbb{R}^m\}_{t \in \mathbb{N}}$, an optimization problem is usually formulated to determine the best $q$:

$$\min_q \int_t \left( \| G(x(t); u(t); q) - z(t) \|^2 + \beta R(q) \right) dt, \tag{2}$$

where $R(q)$ is a regularization term.

Also, when $z(t)$ is seen as a driving signal, the optimization problem can be regarded as a driving-response synchronization problem finding appropriate parameters $q$[14]. Since RC is often simulated on

classical computers, most commonly used RC takes discrete time steps:

$$\begin{cases} x(t+1) = (1 - \gamma)x(t) + \gamma f(Wx(t) + W^{(in)}u(t) + b), \\ y(t) = W^{(out)}x(t), \end{cases} \tag{3}$$

which is a special form of (1), but now with time steps and network parameters more explicitly expressed. In this form, $f$ is usually a component-wise nonlinear activation function (e.g., tanh), the input-to-internal and internal-to-output mappings are encoded by the matrices $W^{(in)}$ and $W^{(out)}$, whereas the internal network is represented by the matrix $W$. The additional parameters $b$ and $\gamma$ are used to ensure that the dynamics of $x$ is bounded, non-diminishing and (ideally)

exhibits rich patterns that enable later extraction. Given some training time series data $\{z(t)\}$ (assumed to be scalar for notational convenience), once the RC system is set up by fixing the choice of $f, \gamma, b, W^{(in)}$ and $W$, the output weight matrix $W^{(out)}$ can be obtained by attempting to minimize a loss function. A commonly used loss function is

$$W^{(out)\top} = \arg\min_w (\|Xw - z\|^2 + \beta \|w\|^2), \qquad (4)$$

where $X = (x(1)^\top, x(2)^\top, \ldots, x(T)^\top)^\top$, $z = (z(1), z(2), \ldots, z(T))^\top$ and $\beta \in [0, 1]$ is a prescribed parameter. This problem is in a special form of Tikhonov regularization and yields an explicit solution $W^{(out)\top} = \left(X^\top X + \beta^2 I\right)^{-1} X^\top z$.

## Common RC designs

Designing is a crucial step for acquiring a powerful RC network. There are still no complete instructions on how to design optimal RC networks based on various necessities. With the unified forms Eqs. (1) and (2) in mind, a standard RC system as initially proposed contains everything random and fixed including the input and internal matrices $W^{(in)}$ and $W$, leaving the choice of parameters $\gamma$ and $\beta$ according to some heuristic rules. Based on this default setting, we show how different RC designs can generally be interpreted as optimizing in one and/or multiple parts along the following directions. Firstly, in RC coupling parameter search, with the goal of selecting a good and potentially optimal coupling parameter $\gamma$ to maintain the RC dynamics bounded and produces rich pattern that allow for the internal states to form a signal bases that can later be combined to approximate the desired series $\{z(t)\}$. Empirical studies have shown that $\gamma$ chosen so that the system is around the *edge of chaos*[15] typically produces the best outcome, which is supported by a necessary but not sufficient condition - imposed on the largest singular value of the effective stability matrix $W_\gamma = (1 - \gamma) + \gamma W$. Then, in RC output training, whose design commonly amounts to two aspects. One is to determine the right optimization objective, for instance the one in Eq. (4) with common generalizations include to change the norms used in the objective in particular the term $\|w\|$ to enforce sparsity or to impose additional prior information by changing $\beta\|w\|$ into $\|Lw\|$ with some matrix $L$ encoding the prior information. On the other hand, (upon choice of the objective) to further determine the parameter, e.g., $\beta$ as in Eq. (4). Although there is no general theoretically guaranteed optimal choice, several common methods can be utilized, e.g., cross-validation techniques that had been well-developed in the literature of computational inverse problems. RC network design is crucial to determine the dynamic characteristics. With the goal of determining a good internal coupling network $W$. This has received much attention and has attracted many novel proposals, which include structured graphs with random as well as non-random weights[16,17], and networks that are layered and deep or hierarchically coupled[18–20]. Furthermore, sometimes those designs are themselves coupled with the way the input and output parts of the system are used, for example in solving partial differential equations (PDEs)[21,22] or representing the dynamics of multivariate time series[23]. Finally, as for RC input design, although received relatively little attention until recently, it turns out that the input part of an RC can play very important roles in the system's performance. Here input design is generally interpreted to include not only the design of the input coupling matrix $W^{(in)}$ but also potentially some (non)linear transformation on the input $u(t)$ and/or target variable $z(t)$ prior to setting up the rest of the RC system. The so-called next-generation RC (NG-RC) is one such example[24], showing great potential of input design in improving the data efficiency (less data required to train) of an RC.

In addition to the separate designs of the individual parts of an RC, the novel concept of neural architecture search (NAS) has motivated the research of hyperparmeter optimization[25] and Automated RC design to (optimally) design an RC system for not just one, but an entire class of problems and ask what might be the best RC architecture - including its input and internal coupling dynamics and training objective[25,26], for instance using Bayesian optimization[27]. Furthermore, nonlinear functions beyond the component-wise $f = \tanh$ are often encountered in experimental settings and an active line of research is to explore new types of nonlinear dynamics such as electro-optic phase-delay dynamics[28–30], optical scattering[31,32], dynamic memristors[33–36], enlarged memory capacity in chaotic dynamics[37], solitons[38] and quantum states[39,40].

## Mathematical theory behind RC

The fundamental questions of exactly why, when and how RC learns a general dynamical process are important mathematical questions whose answers are expected to provide guidelines for the practical design and implementation of RC systems. These lines of queries have led to a number of important analytical results which we classify into four categories.

The first category of work focuses on the echo state property (ESP): Equivalent to state contracting, state forgetting, and input forgetting - refers to RC networks whose asymptotic states $x(t \to \infty)$ depends only on the input sequence and not on the initial network states. This property leads to a continuity property of the system known as the fading memory property where current state of the system mostly depends on near-term history and not long past[11]. Ref. 11 considers RC network with sigmoid nonlinearity and unit output function and showed that if the largest singular value of the weight matrix $W$ is less than one then the system has ESP, and if the spectral radius of $W$ is larger than one then the system is asymptotically unstable and thus cannot has ESP. Tighter bounds were subsequently derived in[41]. In particular, the spectral radius condition provides a practical way of ruling out bad RCs and can be seen a necessary condition for RC to properly function.

The second category is about memory capacity. Defined by the summation of delay linear correlations of the input sequence and output states, was shown to not exceed $N$ for under iid input stream[42], can be approached with arbitrary precision using simple linear cyclic reservoirs[16], and can be improved using the time delays in the reservoir neurons[43].

Universal approximation theorems can be regarded as a single category. Prior to the research of RC, universal representation theorems by Boyd and Chua showed that any time-invariant continuous nonlinear operator can be approximated either by a Volterra series or alternatively by a linear dynamical system with nonlinear readout[44]. RC's representation power has attracted significant recent interest: ESNs are shown to be universally approximating for discrete-time fading memory processes that are uniformly bounded[45] and further that the approximating family can be associated with networks with ESP and fading memory[46]. For discrete-time stochastic inputs, linear reservoir systems with either polynomial or neural network readout maps are universal and so are ESNs with linear outputs under further exponential moment constraints imposed on the input process[47]. For structurally stable systems, they can be approximated (upon topological conjugacy) by a sufficiently large ESN[48]. In particular, ESNs whose output states are trained with Tikhonov regularization are shown to approximate ergodic dynamical systems[49]. Also rigorously, the dynamics of RC is validated as a higher-dimensional embedding of the input nonlinear dynamics[43]. In addition, explicit error bounds are derived for ESNs and general RCs with ESP and fading memory properties under input sequences with given dependency structures[50]. Finally, according to conventional and generalized embedding theories, the RCs with time delays are established with significantly-reduced network sizes, and sometimes can achieve dynamics reconstruction even in the reservoir with a single neuron[43].

The last category includes research about linear versus nonlinear transformations and next-generation RC. Focusing on linear reservoirs

## BOX 3
# Schematic diagram of physical reservoir computing

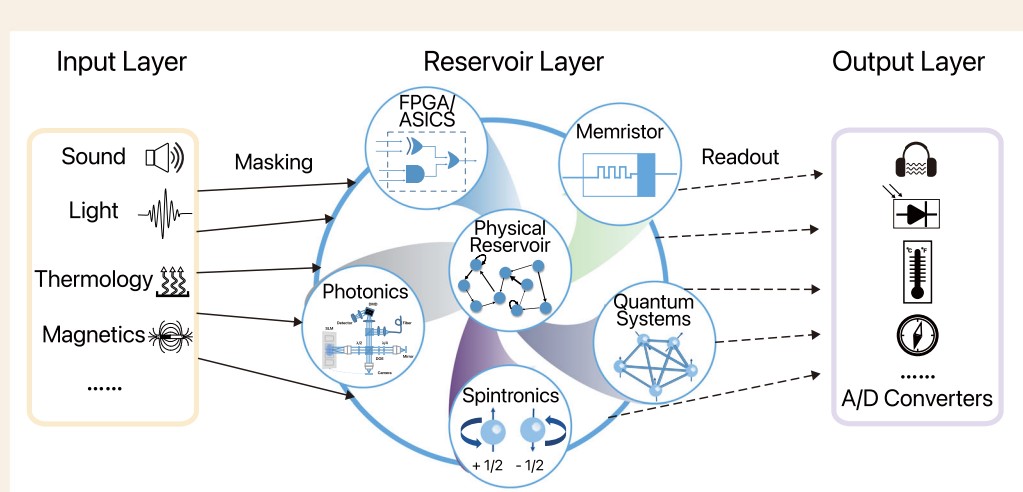

The basic process of physical reservoir computing includes preprocessing external input signal, mapping input signals to a high-dimensional space within a physical system, measuring the physical reservoir state, and finally post-processing to generate output data. Generally, it is necessary to pre-process the raw input data such as sound, light or other physical signals, converting it into a new form that can communicate with physical devices with appropriate scaling. For time multiplexed signals, this step is often referred to as masking. Pre-processed signals can trigger complex dynamic processes in nonlinear physical systems. At this point, the reservoir state can be understood as a high-dimensional distribution of physical variables within the system, and the connection of reservoirs and nonlinear mapping of nodes are determined by the actual physical system. While at the output layer, it is usually necessary to find a suitable measurable physical quantity to reflect changes in internal state, and then convert the corresponding physical signals into electrical domain for the next online or offline training.

(possibly upon pre-transformations of the input states), recent work showed that the output states of an RC can be expressed in terms of a controllability matrix together with the network encoded inputs[17]. Moreover, a simplified class of RCs are shown to be equivalent to general vector autoregressive (VAR) processes[51] - with possible nonlinear basis expansions it forms theoretical foundations for the recently coined concept of next-generation RC[24].

Research of how to design RC architectures, how to train them and why they work have, over the past two decades following the pioneering works of Jaeger and Maass, led to much evolved view of the capabilities as well as limitations of the RC framework for learning. On the one hand, simulation and numerical research has produced many new network architectures improving the performance of RC beyond purely random connections; future works can either adopt a *one-fits-all* approach to investigate very large random RCs or perhaps more likely to follow the concept of domain-specific architecture (DSA)[52] to explore structured classes of RCs that achieve optimal performance for particular types of applications, with Bayesian optimization[26,27] and NAS as powerful tools of investigation[53]. On the other hand, for a long time only few theoretical guidelines based on ESP were available for practical design of RCs; more recently several important theoretical

discoveries were made establishing universal approximation theorems of RC - those results, although not yet directly useful for constructing optimal RCs, may nevertheless boost confidence and stimulate explicitly ideas of designing and even optimizing RCs for learning. In particular, despite having randomly assigned weights that are not trained, RC models are nevertheless shown to possess strong representation power with rigorous theoretical guarantees.

## Physical design of RC systems: from integrated circuits to silicon photonics
To archive a controllable nonlinear high-dimensional system with short-term memory, some specific physical systems with nonlinear dynamic characteristics can be used to implement reservoirs (see Box 3), where network connections are determined by the physical interactions. As the development of integration technology for electrical and optical component, the computational efficiency can be greatly improved compared to traditional Boolean logic methods. The implementation of physical reservoir is similar to the software approach, but slightly different.

In recent years, there has been extensive research on designing and realizing RC using physical systems. A detailed review can be

**Table 1 | Comparison between typical physical RC implementation methods**

| Reservoir schemes | Physical system | Node nonlinearity | Number of Nodes | Operating Speed[1] | Energy Efficiency | Program-Ability[2] | I/O Types | Refs. |
|---|---|---|---|---|---|---|---|---|
| Integrated circuits | FPGAs | Nonlinear circuits based on standard Boolean Gates | $10 \sim 10^3$ | ~MHz | ~mW | High | Electric-Electric | 71,130,131 |
| | ASICs | | $10 \sim 10^3$ | ~MHz | ~μW | Low | Electric-Electric | 73,74,132,133 |
| Novel nonlinear circuits | Memristor | Non-static resistance | $10 \sim 10^2$ | ~MHz | ~mW | High | Electric-Electric | 33,35,36,78 |
| | Spintronics | Nonlinear spin-electrical dynamics | $10 \sim 10^2$ | ~MHz | ~μW | Medium | Electric-Electric | 65,79,134 |
| Photonic systems | Silicon photonics | Photoelectric effect; SOA; saturable absorption; laser dynamics | $\sim 10^2$ | ~THz | ~mW | High | Electric-Optical-Electric | 66,86 |
| | Fiber optics | | $\sim 10^2$ | ~GHz | μW ~ mW | Medium | Electric-Optical-Electric | 58,68,135 |
| | Free space optics | | $\sim 10^4$ | ~GHz | μW ~ mW | High | Electric-Optical-Electric | 31,32,127 |

Note 1. The operating speed of the system is influenced by data preprocessing, A/D-D/A conversion, node nonlinear response time and other factors. Ideally, the operating speed can reach the response time limit of the nonlinear nodes.
Note 2. Programmability is determined by if a reservoir can be trained and modified.

found in reference[54]. Physical reservoirs can be roughly divided into three types based on their topological structure: discrete physical nodes reservoir, single-node reservoirs with delayed feedback and continuous medium type reservoirs. Discrete physical nodes reservoir is composed of interacting nonlinear components, such as memristors[35], spintronics[55], oscillators[56], optical nodes[32], etc. The nodes form a coupling network through real physical connections. They can be simply enlarged by increasing the number of network elements to obtain higher dimensions. Single-node reservoir is composed of a single nonlinear node and a time delay loop, which can transform the input signal into a virtual high-dimensional space through time division multiplexing using single nonlinear physical nodes, such as analog circuits[57], lasers[58], etc. This type of reservoir avoids the problem of large-scale interconnection, making it more hardware friendly. However, designing and implementing appropriate delayed feedback loops is not a simple task. Continuous-medium reservoir mainly utilizes the physical phenomena of various waves in a continuous medium, such as fluid and elastic media. This type of physical system can utilize the physical properties of waves, such as interference, resonance, and synchronization, to achieve extremely efficient physical RC[59]. In terms of specific physical schemes, there are also physical reservoirs implemented by mechanical[60], biological[61], quantum systems[39] and superconductors[62]. In this article, we mainly focus on comparing various physical implementation solutions in terms of integration, power consumption, processing speed, and programmability, as shown in Table 1. Typical high-performance physical reservoirs include traditional electronic schemes represented by Boolean logic circuits such as FPGA[63] and ASICs[64]; Non-Von Neumann electrical reservoir scheme represented by memristor[33] and spintronics[65] devices; And photonic schemes represented by silicon photonics[66], fiber optics[29,67,68] and free-space optics[69].

In principle, existing morphological circuits, such as FPGAs and ASICs, can be implemented as an electronic reservoir. With its bit-level fine-grained customized structure, parallel computing ability, and efficient energy consumption, FPGAs exhibit unique advantages in deep learning applications. Using FPGAs for reservoir computing is also advantageous, as sparse connections in the reservoir model allow for simple routing techniques that match FPGA requirements. Currently, several FPGA methods have been proposed[70–72]. In addition, considering the high programming requirements of FPGAs, people have proposed the implementation of RC algorithm using Application Specific Integrated Circuits (ASICs)[73], which can help improve chip performance and power consumption ratio. The disadvantage of ASIC-based RC is that circuit design customization leads to relatively long development cycles, inability to scale, and high costs. But research in this area is also actively advancing[74,75].

Besides the electric reservoir that is based on Boolean logic and von-Neumann architecture, people have been pursuing higher efficiency and lower energy consumption methods. For the reservoir model, the nonlinear analog electronic circuit can be used to directly build the reservoir model, such as the Mackey - Glass circuit[76]. Based on nonlinear electronic circuits, a single electric node, such as a memristor, or a spintronic device, with delay lines that can be constructed and combined with other digital hardware components for pre-processing and post-processing[77]. The memristor has the dimension of resistance, but its resistance value is determined by the charge flowing through it. It functions as a memory, and can generate rich reservoir states under an appropriate time division multiplexing mechanism[35]. In addition, it is also possible to construct 2D/3D memristor crossbar arrays and encode matrix elements into the embedded memristor conductance[78]. This programming can be accomplished using voltage pulses with minimal energy required. On the other hand, micro/nano spin electronic devices constructed using electron spin degrees of freedom can exhibit the physical properties of tiny magnets and can be used to simulate synaptic behavior in biological nerves[65]. At present,

people have proposed several reservoir schemes based on the physical phenomena related to spintronics[79].

On the other hand, development of photonic technology has brought hope for ultra-high speed and low energy consumption hardware systems, especially for neural network training[80]. Optical systems have significant advantages over traditional microelectronic technologies in terms of high bandwidth, low latency, and low energy consumption. Reservoir networks based on optical systems have also made significant progress[81], such as multi-scattering nodes in free space[32], single nonlinear nodes with fiber loop[58], and integrated on-chip reservoirs[66]. The free-space reservoir is generally achieved using spatial optics and scattering media, such as diffractive optical elements (DOE), to achieve coupling between spatial optical nodes. Interconnection between neurons in the reservoir are realized through complex scattering processes[32]. Single nonlinear optical nodes, such as semiconductor optical amplifiers (SOAs), saturable absorbers (SESAM), as well as semiconductor lasers can form optical reservoirs with special fiber loop designs[81]. Integrated on-chip optical reservoir are often archived by interaction between nonlinear micro/nano optical devices, such as micro-rings[81]. Unlike the fiber delay loop architecture, utilizing multiple on chip nonlinear optical nodes makes it more convenient to take advantage of optical parallel computing.

Comparatively speaking, reservoir schemes based on FPGAs and ASICs can greatly improve the computing speed and power consumption compared with the general CPU electronic architecture, due to its non-Von Neumann/in-memory nature of the computing. Besides, there is no need for photoelectric conversion at either the input or output ends, making it convenient in data scaling and processing. However, the computing efficiency is close to the theoretical limit. For electrical non-Von Neumann architectures, such as memristors, more efficient computation can be realized theoretically, but due to their analog nature, it is usually difficult to realize ideal nonlinear mappings and high-precision matrix calculations, and the integration and stability of such devices also need to be improved. As for spintronics reservoirs, so far, most studies have only explored nanomagnetic RC in simulations, and it also faces the scalability problem similar to memristor. For optical reservoir schemes, the low delay and low energy consumption characteristics of optical devices are generally only reflected in the reservoir layer. Currently, most schemes require photoelectric conversion in data preprocessing and post-processing, and the response time of the system is essentially limited by photodetectors and the time delay of electronic control circuits. At the same time, optical processing errors and the power consumption of external auxiliary devices also pose strict limitations on the scale of the system. So it seems there is currently no solution that can be said to be the best. In the short term, electronic solutions such as FPGAs do not require photoelectric conversion in the input and output processes, and are measurement friendly, thus having advantages in hardware implementation. However, considering issues such as power consumption and latency, specialized photonic reservoirs will have more advantages in the future. Perhaps utilizing their respective advantages comprehensively and adopting a heterogeneous integration solution is a feasible path.

## Application benchmarks of RC

Applications of RC are quite diverse and can be mainly divided into several categories: signal classification (e.g., spoken digit recognition), time series prediction (e.g., chaos prediction such as in the Mackey-Glass dynamics), control of system dynamics (e.g., learning to control robots in real-time) and PDE computations (e.g., fast simulation of Kuramoto-Sivashinsky equations), which we discuss below respectively.

In signal classification tasks, the input of RC are usually broadly-interpreted (physical) signals such as audio, image or temporal waves. The target output are the corresponding labels which can be spoken digits[16,28,29,34,35,65,68,82–84], image labels[33,35,85–87], bit-symbols[16,29,68,88–93] and so on. The effectiveness of traditional neural networks in classification tasks has been verified in lots of work. However, dealing with temporal input signal is still a challenge. Compared with traditional neural networks, RC can map temporal signals with multiple timescales to high dimension, encoding these signals with its various internal states. Furthermore, RC network has much less parameters thus requiring less training resources. Therefore, RC can be a good candidate to be utilized in temporal signal classification tasks. The signals are in various types (audio, image or temporal waves), and usually require some preprocessing before injecting to RC network. For example, in the spoken-digit recognition task, the raw signal is first transformed to frequency domain in terms of multiple frequency channels via Lyon's passive ear model, as shown in Fig. 2a. Then the 2-D signals can be directly mapped to the RC network as input $u(t)$ via input mask, or can be transformed to 1-D input sequence $u(t)$ by connecting each row successively. The targets are a vector of size ten corresponding to digit number from 0 to 9. The state-of-the-art of RC currently can reach a word error rate (WER) of 0.4% from memristor chip RC[35], and 0.2% from electronic RC[28].

For time series prediction, RC assumes the role of regression, taking input as a segment of time series up to a certain time and draws predictions for the next (few) time steps. Examples are abundant, including prediction of chaotic dynamics such as Mackey-Glass equations[11,31,34,51,94], Lorenz system[22,26,49,51,94–96], Santa Fe Chaotic time series[16,86,89,94], Ikeda system[94], auto-regressive moving average (NARMA) sequence[16,28,29,93,94,98], Hénon map[16,35,94,98], radar signal[68], language sentence[36], stocks data[61], sea surface temperatures (SST)[99], traffic breakdown[100–102], tool wear detection[96] and wind power[103]. Given a training time series $\{z(t)\}_{t\in\mathbb{Z}}$ and prescribed prediction horizon $\tau$, the input sequence of RC can be defined as $u(t) = z(t)$ while the target output as $y(t) = z(t+\tau)$. (For one-step prediction we use $\tau = 1$.) Once the parameters of RC is learned, it can be used as a predictive model, taking a temporal input and predicts its next steps. In particular, RC trained with one-step prediction can nevertheless be used to make multi-step predictions, in the following way. Suppose that a finite-length time series $\{u(t)\}_{t=1,...,T}$ is provided, we feed it into RC to compute a state $y(t+1)$ as a one-step prediction. We then append this state to the end of the input effectively defining $u(t+1) = y(t+1)$ and through RC to compute a next state $y(t+2)$, and so on and so forth to obtain a series of next steps $y(t+1, t+2, ..., t+h)$. A schematic example of nonlinear time series prediction task is shown in Fig. 2b. In order to realize long-term prediction, there is another training scheme in which the target sequence is inserted periodically. In particular, the input to the RC now comes from its feedback or target sequence alternately, as shown in Fig. 2b (method 2). Compared with the previous case which can be regarded as an offline training scheme, here RC can acquire target data periodically, then retraining and updating the output weights regularly. This is an online training scheme. Since RC has access to target data during its evolution, it can adjust the output weights to prevent the predictive output data from diverging. Therefore, the online training typically yields longer prediction period and better prediction performances.

RC can play important roles in the control of nonlinear dynamical systems[104–109]. In particular, in the model predictive control (MPC) framework, control actions are derived based on a predictive model of the system dynamics. The predictive model is typically linear due to simplicity and low computational cost. RC as an alternative can potentially improves upon the linear prediction without introducing too much additional computational overhead, as shown in Fig. 2c. As a concrete example, in the controlling robot arm movement task[104], the mechanical arm gives input data such as joint arm angles, destination position coordinates and joint arm torques calculated from Lagrangian equation. RC is trained with the targets which are successive joint torques needed to gradually move to the destination. In the testing

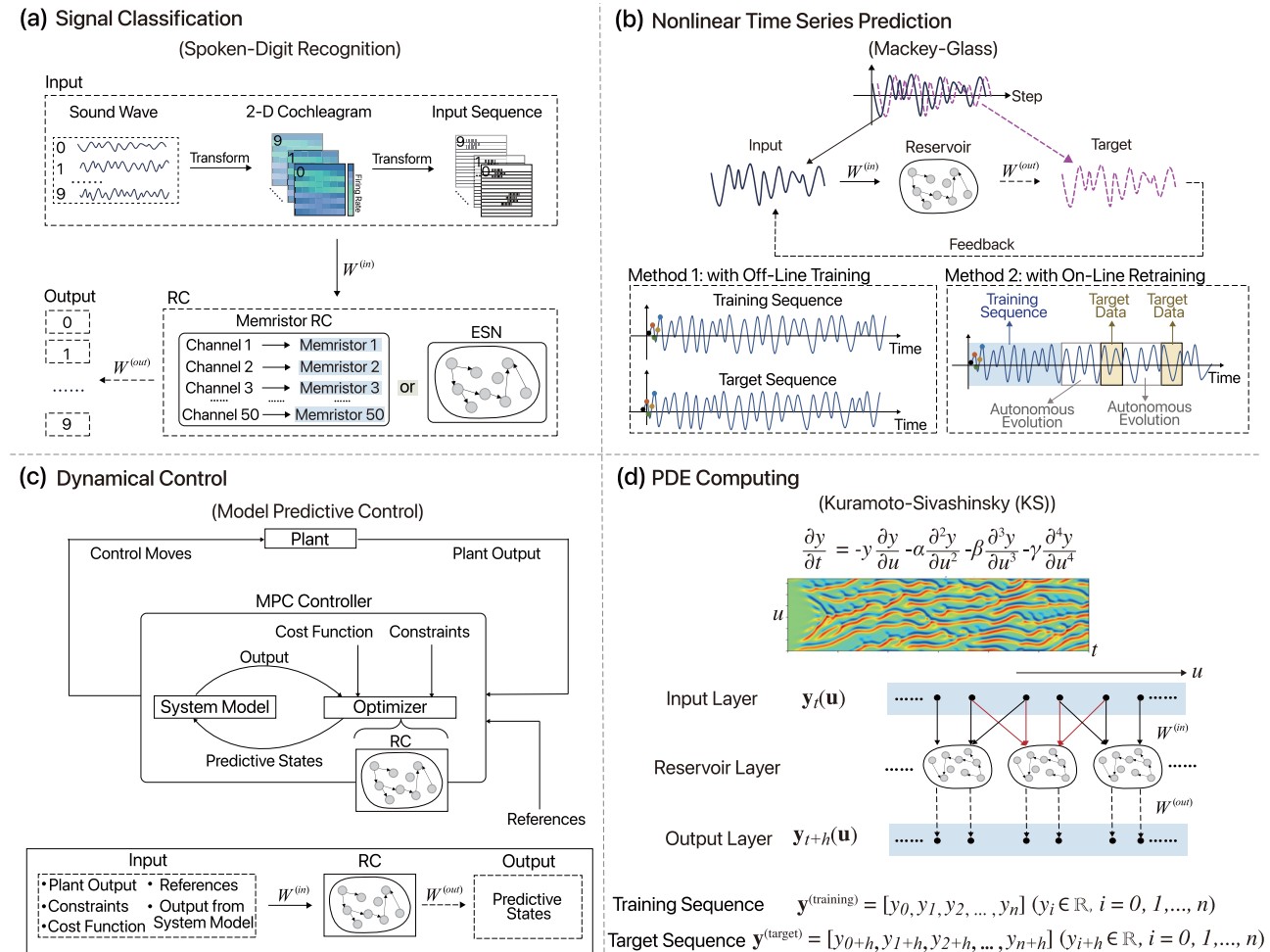

**Fig. 2 | Example applications of RC.** Flow diagrams showing how RC is applied in different types of applications, here referring to as signal classification, nonlinear time series prediction, dynamical control and PDE computing, respectively. **a** RC for spoken-digit recognition[16,28,29,34,35,65,68,82–84], when the targets are a vector of digit numbers corresponding to 0–9. **b** RC for time series prediction with Mackey-Glass equations[11,31,34,51,94] as an example. In method 1 with off-line training, the training sequence starts with the first point (black point), while the target sequence starts with the second one (orange point). In method 2 with on-line retraining, the training

and testing are alternately presented. **c** RC acts as the prediction optimizer in the general model predictive control (MPC)[104–109] framework. Top: The MPC diagram. Bottom: How RC works in the MPC system. **d** RC for PDE computation[21,22,32] with the Kuramoto-Sivashinsky (KS) equations as an example. The hidden layer consists of parallel multiple reservoirs, and each of them deal with part of the input data, while a nonlinear transformation is typically inserted before training the parameters of the readout layer.

phase, as long as the destination is given, the robot arm can evolve by itself to approximate to the target point. Additionally, RC jointly with adaptive feedback control technique can be used to track the unknown and unstable periodic orbits and stabilize them even when the chaotic time series are only available[14].

RC can be applied for scientific computation such as in the numerical solution of PDEs[21,22,32]. For these tasks, RC is typically used to evolve the states of the system toward the temporal direction with a flow diagram shown in Fig. 2d. To decrease the difficulty for a single reservoir to process all inputs and improve the training efficiency, parallel reservoir architecture was proposed[21,22] which allows multiple small reservoirs to deal with different parts of input data. The input vector is split into multiple small groups with each group includes some extra adjacent points serving as extra information provided to the corresponding small reservoirs. The target is the next time step vector of the PDE. Accompanied with a nonlinear readout function, the RC network can learn and evolve Kuramoto-Sivashinsky (KS) equation relatively accurate up to a time length of around 5 Lyapunov times[21].

Overall, RC has demonstrated strong performance across a range of benchmarks and tasks, with ongoing efforts to further

improve results. A summary of trends in RC performances in typical application scenarios is shown in Fig. 3. For example, in spoken digit recognition, WER is reaching near-perfect levels (0.014%)[58]. Similarly, handwritten digit recognition boasts an accuracy of around 97.6%[35]. While RC currently has limitations in action recognition and requires preprocessing, there is potential for future development in expanding recognition abilities and reducing preprocessing needs. In time series prediction, RC excels in chaotic sequences such as Mackey–Glass, Lorenz, and Santa Fe, but real-world data such as weather[110,111], stocks, and wind power show less impressive performance. RC is primarily used in dynamic control for MPC systems, but as system complexity increases, real-time control with greater accuracy and efficiency is necessary. Lastly, RC has been shown to compute PDEs effectively, but practical applications of this ability have yet to be fully realized. Despite attempts and preliminary successes in applying RC to problems endowed with real-world datasets, nearly none of those attempts have led to an industry-level adoption and application. An important reason is that the performance of RC on common tasks such as image classification, audio signal processing have not reached or shown to have the potential to approach the SOTA metrics offered

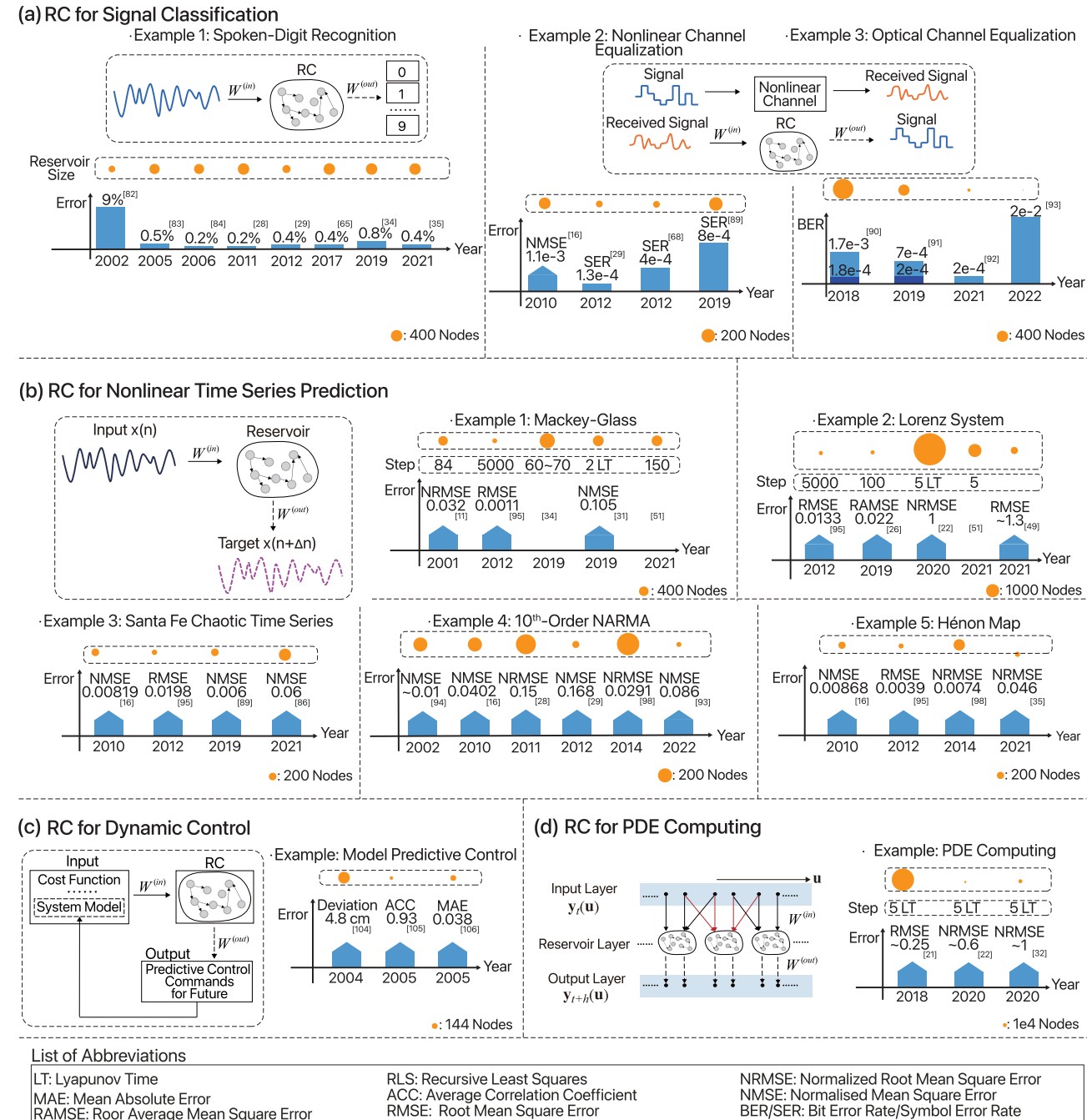

**Fig. 3 | Trends in RC performance in typical application scenarios.** Four kinds of representative scenarios are: **a** signal classification tasks such as spoken-digit recognition, nonlinear channel equation and optical channel equalization; **b** time series prediction such as predicting the dynamics of Mackey-Glass equations,

Lorenz systems as well as and Santa Fe chaotic time series; **c** control tasks and **d** PDE computation. Thick, up-pointing arrows in the panels denote error values that are not directly comparable with other works.

by deep-learning based methods. Given that theoretically RC has universal approximation capacities just as general neural networks, in principle nothing seems to be holding back RC models to push the frontiers of most challenging AI tasks, and this should be a main goal of the entire RC community.

## Opportunities and technical challenges for future development of RC

We expect that research in RC can play important roles in several important application domains, which we discuss as follows. As technology continues to rapidly advance, there is an increasing demand to develop intelligent information processing systems

that are both dynamic and lightweight, yet widely deployable at low cost. According to estimates, by the years 2030–2035, both wireless and optical communication will usher in the sixth generation (6G/F6G), providing connections for tens of billions of devices and multi-billion users[112,113]. It is also expected that global data centers will have a throughput of trillions of GB and require over 200 terawatt hours of power consumption[114]. Furthermore, tens and hundreds of millions of robots are set to enter our daily lives to improve labor efficiency at a low cost[115]. Virtual reality and Metaverse rely heavily on real-time simulation of the physical world[116]. These major applications require a large number of capabilities, including accurate recognition of dynamic

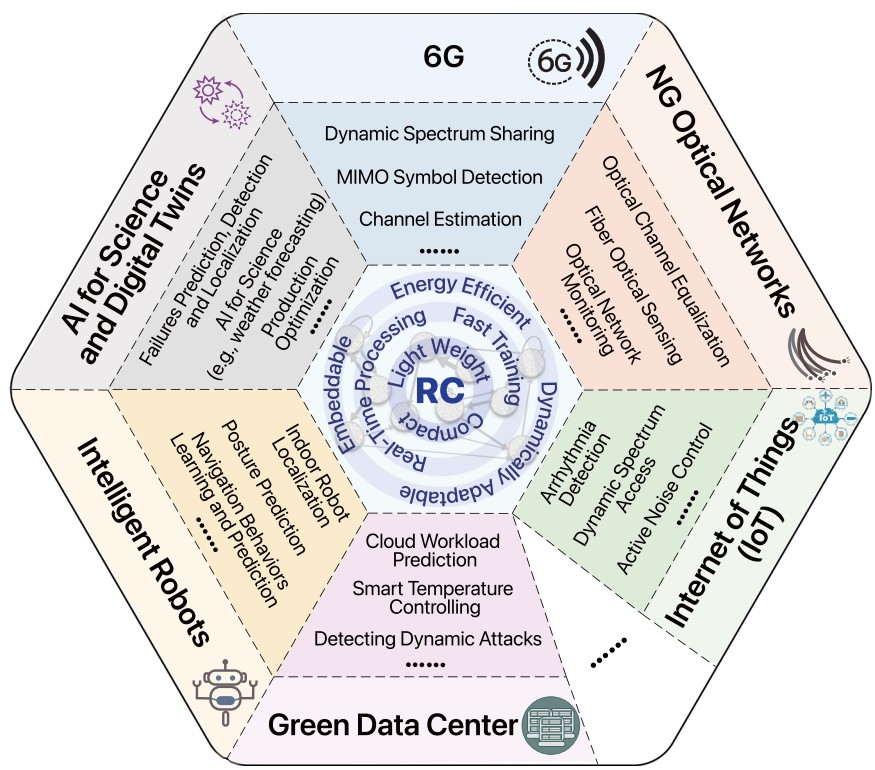

**Fig. 4 | Application domains in which RC potentially can play important roles.** Each domain corresponds to three specific example application scenarios. Six domains are 6G[136–140], Next Generation (NG) Optical Networks[92,93,141–143], Internet of Things (IoT)[61,144,145], Green Data Center[120,146,147], Intelligent Robots[148–150] and AI for Science[151–157] and Digital Twins[99,103,158–161].

uncertainty information, fast prediction and computation, and dynamic control, all of which can be provided by RC systems, as shown in Fig. 4. As a result, we expect that RC research will play a critical role in several important application domains, as we will discuss below.

## 6G

*Opportunities*. It is predicted that by 2030, wireless communication will advance to its sixth generation, commonly referred to as 6G. The main goal for 6G is to enhance important indicators such as transmission speed, coverage density, time delay, and reliability by 10 to 100 times compared to 5G. This would provide a never-before-seen connection experience across a wider area for numerous devices[112,113]. *Challenges*. In order to realize the beyond-5G vision, several technical challenges need to be addressed. The most crucial one is achieving low-latency, high-reliability network connections for complex channel environments and providing deterministic communication guarantees. The key to this is active signal processing through predicting potential changes in the channel based on the perception of the environment. This requires overturning traditional passive waveform design and channel coding, and instead, relying heavily on active sensing, accurate prediction, and dynamic optimization of complex channels to systematically optimize channel capacity. To address these technical challenges while maintaining a lightweight deployment cost, RC can play a significant role. For example, essential modules such as waveform optimization and decoding can greatly benefit from accurate identification and dynamic estimation of channel state information integrated sensing and communication. These modules can also be further improved by transforming from responsive to predictive channel estimation. Finally, real-time channel optimization, such as using RIS techniques, would require fast and adaptive control of potentially high-dimensional dynamics. Due to its compact and lightweight network structure, rich functional

interfaces, and low-complexity training and computing nature, RC is expected to become a key technology base for edge-side information processing.

## Next-generation optical networks

*Opportunities*. Optical fiber communication is often regarded as one of the most significant scientific advancements of the 20th century, as noted by Charles Kuen Kao, the Nobel Prize winner in Physics[117]. The optical network derived from optical fiber technology has become a fundamental infrastructure that supports the modern information society, processing more than 95% of network traffic. The next-generation optical fiber communication network aims to achieve a *Fiber to Everywhere* vision[118,119], featuring ultra-high bandwidth (up to 800G-1.6Tbps transmission capacity per fiber), all-optical connectivity (establishing an all-optical network with ultra-low power consumption and extending fibers to deeper indoor settings), and an ultimate experience (zero packet loss, no sense of delay, and ultra-reliability). *Challenges*. To attain such a significant vision, significant technological advancements must be made in areas such as all-optical signal processing, system optimization, and uncertainty control. These technical challenges can benefit from new theories, algorithms, and system architectures of RC. For instance, a silicon photonics integrated RC system, functioning as a photonic neural network, can achieve end-to-end optical domain signal processing with negligible power consumption and time delay in principle, without relying on electro-optical/optical conversion. As a result, it has the potential to become a key technology in future all-optical networks. Additionally, adjusting the internal structure of the optical fiber can enable the enhancement of capacity by searching complex and diverse structures, which can benefit from the effective and automated modeling of the channel with RC. This approach transforms the original black-box optimization of the system into the white-box optimization of

the RC's output layer, likely able to improve the optimization efficiency. In terms of low-latency and reliability assurance at the optical network level, RC research can play a critical role in link failure prediction early warning, fault localization, and dynamical control. Due to the compact design of RC, embedded devices can perform intelligent processing tasks as a natural part of the network system, without requiring a centralized power center.

### Internet of Things (IoT)

*Opportunities*. In comparison to traditional communication and interconnection services for computers and mobile phones, the Internet of Things (IoT) caters to a wider range of devices with broader coverage, posing several new technological challenges. With IoT, the quantity and types of objects served are significantly higher, including smart temperature and light control[120,121], open-space noise cancellation[122], air quality monitoring[123], among others, all of which are key features of smart homes. Communication technologies used to realize the interconnection of these devices are diverse, including Bluetooth, NFC, visible light, RFID, WiFi, ZigBee, and so on. *Challenges*. Unlike high-end devices such as computers and mobile phones, a vast majority of IoT connected devices cannot rely on energy-hungry integrated chip technology to achieve advanced computing performance due to power consumption and volume limitations. Consequently, IoT end-side systems must utilize low-power, programmable techniques to achieve adaptive perception and computing necessary for edge intelligence. The lightweight and dynamically controllable nature of such requirements make RC systems particularly advantageous over large AI models. With the success of domain-specific chips for audio and video processing, there is expected to be significant demand for embedded smart chips in the IoT field, which will open up new opportunities for the application of RC research.

### Green data centers

*Opportunities*. Data centers have become an essential infrastructure for the new generation of information society due to the substantial increase in demand for massive computing and data storage. It is estimated that by 2030, global data centers will process a trillion GB of data every day, and their power consumption is expected to account for over 60% of total power generation. However, the large amount of electricity consumption and heat emissions required to operate these centers have a significant impact on the environment. Therefore, the design and development of new generation green data centers with low energy consumption and high reliability are crucial for the sustainable development of society. *Challenges*. The realization of low-energy data centers relies on numerous technological breakthroughs. Energy consumption in data transfer accounts for a significant proportion, with optical modules playing a central role. Therefore, achieving low energy consumption requires reducing the energy consumption of optical modules. One promising approach is to implement all-optical signal processing based on the integrated silicon photonics on-chip RC system. Additionally, data centers comprise many components that form an extremely complex dynamic system. Maintaining the system's normal operation at the least possible cost of energy consumption, such as keeping the overall temperature stable at a low-range, can be viewed as an optimal control problem. A potential solution to this problem is through data-driven models with physical priors, which combines a structured model derived from the connection relationship and functions of physical equipment and data-driven methods to build a dynamic control framework. By monitoring and adjusting the parameter configuration of each module of the system in real-time, this framework can achieve the optimal operating status and energy consumption cost. RC has the potential to play a crucial role in this approach.

### Intelligent robots

*Opportunities*. Robots are becoming increasingly important in today's information society due to their ability to take many forms, including intelligent physical manifestations. One example of this is large-scale commercial sweeping robots used in smart homes[115], which have replaced traditional manual operations in various scenarios, improving both production efficiency and living standards. With advances in technology, more types of intelligent robots are expected to emerge over the next decade, capable of completing complicated tasks through autonomous perception, calculation, optimization, and control in complex environments like failure detection, medical diagnosis, and search-and-rescue operations. Biological intelligence serves as inspiration for achieving robot intelligence, which relies on three key elements: real-time intensive information collection and perception capabilities (made possible by technologies such as flexible sensing, electronic skin, and multi-dimensional environment modeling), fast information processing capabilities (enabled by technologies like decision-making optimization and dynamic control), and physical control capabilities (facilitated by nonlinear modeling and electro-mechanical control). *Challenges*. Due to physical constraints such as battery capacity and deployment environment uncertainty, the core modules supporting robot intelligence are expected to be embedded in the physical entity of the robot in an offline manner rather than relying on cloud and network capabilities to provide potential large model capabilities. Similar to the IoT scenario, machine learning that is widely relied on in robot intelligence must have the characteristics of miniaturization, low energy consumption, and easy deployment, while requiring the ability to recognize, predict, calculate, and control dynamic processes. This presents an excellent application field for RC systems to play a role. In MPC, since the role of RC merely replaces a linear predictor the overall controller architecture remains transparent and intact. In principle, it is possible to adopt RC for general controller design beyond usage in the MPC framework, e.g., directly learning control rules from data together with (some) prior model knowledge. However, the main challenge would be to pose theoretical guarantees on error and convergence neither of which have been resolved by existing works of RC.

### AI for science and digital twins

*Opportunities*. To fully realize the ongoing information revolution, it is essential to rethink and reshape crucial aspects of industrial manufacturing through the innovative framework of AI for science and digital twins. This involves achieving full perception and precise control of physical systems through interactions and iterative feedback between digital models and entities in the physical world. Essentially, digital twins establish a synchronous relationship between physical systems and their digital representations. Using this synchronous function, simulations can be run in the digital world, and optimized designs can repeatedly and iteratively be imported into the physical system, ultimately leading to optimization and control. For systems with clear and complete physical mechanisms, synchronization models that digital twins rely on are usually sets of ODEs/PDEs. For example, simulating full three-dimensional turbulence, weather forecasting, laser dynamics, etc. Preliminary studies suggest that reservoir computing can be used to reduce the computational resources required for these expensive simulations. Arcomano et al.[111] developed a low-resolution global prediction model based on reservoir computing and investigated the applicability of RC in weather forecasting. They demonstrated that a parallel ML model based on RC can predict the global atmospheric state in the same grid format as the numerical (physics-based) global weather forecast model. They also found that the current version of the ML model has potential in short-term weather forecasting. They further discovered that when full-state dynamics are available for training, RC outperforms the time-based backpropagation through time (BPTT) method in terms of prediction

performance and capturing long-term statistical data while requiring less training time. *Challenges*. Calculations of these physics-inferred equations can be challenging. In more complex industrial applications, multiple coupling modules are often present, and interactions between the system and the open environment cannot be fully described by physical mechanisms or mathematical functions. Therefore, it is necessary to consider fast calculation techniques, but also find ways to build synchronization models for non-white-box complex dynamic systems. Mathematical modeling of fusion between physical mechanisms and data-driven techniques has been significantly developed in the past decade. For instance, Physics-inspired Neural Networks (PINN) embed the structure and form of physical equations into neural network loss functions, which guides the neural network to approximate provided physics equations during parameter training[124]. Another type of physics-inspired computing system, RC, inherently provides an embedding method of the mechanism model, which is expected to provide a powerful supplement to the solver for basic physical models of industrial simulation, focusing on offering a dynamic modeling framework for the fusion of mechanisms and data. However, for reduced-order data, large-scale RC models may be unstable and more likely to exhibit bias than the BPTT algorithm. In another example of research on nonlinear laser dynamics, the authors found that RC methods have simpler training mechanisms and can reduce training time compared to deep neural networks[125]. For practical problems involving complex nonlinear physical processes, we have reason to believe that RC methods may provide us with solutions for computational acceleration.

## Outlook

In summary, although RC has the potential for large-scale application in terms of functions, in order to truly solve the technical problems in the above-mentioned various major applications, there are still many key challenges in the existing RC system in various aspects. For example, in theoretical research, although the universal approximation theory of RC has advanced significantly in recent years, most of the theoretical results focus on existence proofs and lack structural design. Hence, the current approximation theory has not yet played an important guiding role in RC network architecture design, training methods, etc., nor can it quantitatively evaluate the approximation potential of a specific RC scheme for dynamic systems or time series. An important reason to further advance the mathematical theory of RC is for data-driven control applications. In most of those applications, rigorous theory on control error and convergence are necessary for the corresponding controller to be considered usable in an industrial setting. However, so far very little work has been done to address these important problems. As for algorithmic challenges, most industrial applications do not require a universal approximator, but in the same field, the approximation model needs to be generalizable. Existing RC research has very little exploration in domain-specific architecture optimization. Problems in the industrial field are divided into scenarios and categories. Therefore, it is important to construct general-purpose RC models possibly by means of architecture search. In addition, leaving aside the practicality of RC for the time being, past research has turned its advantages into constraints, such as small size, simple training, and so on. However, how strong is RC's learning ability (whether there is an RC architecture that can compare with GPT's ability), it is still unknown.

At the experimental level, there are still some gaps when mapping RC models to physical systems. The first is timescale problem of physical substrate RC: Matching the timescales between the computational challenge and the internal dynamics of the physical RC substrate is a key issue in reservoir computing. If the timescale of the problem is much faster than the response time of the physical system, the response of the reservoir will be too small or the fading memory of the reservoir will not be properly utilized, rendering the physical

reservoir computing system ineffective. One intuitive solution is to adjust the physical parameters of the reservoir to match the timescale of the computational problem. This poses high requirements for the design of RC network structures and training algorithms. Using other technologies such as super-resolution and compressive sensing to overcome the resolution problem of single-point measurement and processing in RC systems may be a viable solution. The second is the real-time data processing problem: One of the significant advantages of reservoir computing is lightweight and fast computation. However, in practical physical systems, it is often unrealistic to sample and store a large number of node responses to a certain input due to limitations such as sampling bandwidth, storage depth and bandwidth, or their combinations. It is simply not feasible in many cases to probe a system with a large number of probes (10s–1000s) interfaced with AD converters. In addition to these practical challenges, hardware drift often requires regular repetition of calibration procedures, hence it cannot be a one-of optimization. Furthermore, data preprocessing and post-processing also limit the overall computational speed of the physical RC system. One approach to address this issue is to use hardware-based readout instead of software-based readout[126–129].

Moving forward, it is crucial that we thoroughly explore the potential of intelligent learning machines based on dynamical systems. In the realm of theoretical and algorithmic research, it is necessary to continuously push the boundaries of performance and offer guidance for experimental design. Reservoir computing (RC) research can take root in theory and algorithms, with experiments serving as approximations to theoretical and algorithmic results. However, one disadvantage of this approach is that it can be challenging to identify equivalent devices in experiments that can achieve the nonlinear properties of RC in theory, which can lead to reduced accuracy. Alternatively, researchers can focus on building physical RC system as the ultimate goal, which requires close collaboration between theoretical and experimental teams to optimize the system jointly. This approach has the advantage of considering physical constraints and application characteristics when designing algorithms, making it more likely to achieve better solutions at the implementation level. This also raises the bar for interdisciplinary research, as participants will need to possess cross-disciplinary communication skills and knowledge, along with an openness towards multi-module complex coupling optimization.

Looking ahead, unlocking the full potential of RC and neuromorphic computing in general is critical yet challenging. In fact, this goes beyond just putting out open-source codes or solve a few specific problems. Innovative ideas and interdisciplinary research formats are much needed. As concrete suggestions, researchers of the applied mathematics and nonlinear dynamics communities who have been the main players in RC will need to get close(r) to the mainstream AI applications and try to develop next-generation RC systems to compete in these scenarios where the value of application has been established and recognized by the industry. A good starting point can be open-source tasks and datasets such as Kaggle, and more generally to directly partner with industrial research labs to put RC into real applications. On the other hand, raising awareness of the (potential) utility of RC requires attracting interest from researchers and decision-makers who are traditionally outside of the field. For instance, themed conferences and workshops may be organized to foster such discussions among scientists and researchers from diverse fields across academia and industry. Despite the many challenges, with persistence and innovations a new and future paradigm of intelligent learning and computing may possibly emerge from the works of RC and neuromorphic computing.

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

## Acknowledgements
W.L. is supported by the National Natural Science Foundation of China (No. 11925103) and by the STCSM (Nos. 22JC1402500, 22JC1401402, and 2021SHZDZX0103). P.B. is supported by the EU H2020 program under grant agreements 871330 (NEoteRIC), 101017237 (PHOENICS), 101098717 (Respite), 101046329 (NEHO), 101070238 (Neuropuls), 101070195 (Prometheus); the Flemish FWO project G006020N and the Belgian EOS project G0H1422N.

## Author contributions
J.S., C.H. and M.Y. initiated the paper and developed its outline. J.S., C.H. and M.Y. wrote the first draft. P.B., P.T. and W.L. contributed substantially during the preparation of the manuscript. All authors approved the submission.

## Competing interests
The authors declare no competing interests.
