## [Peer Review File · Nature Communications]

Reviewers' comments:

Reviewer #1 (Remarks to the Author):

The authors have prepared a concise summary of reservoir computing, tracing its history, success stories, and pointing out outstanding problems and likely future high-impact applications. This will likely be a nice article for researchers in the reservoir computing field.

But it is not clear to me that the article will broaden the appeal of reservoir computers outside of the immediate community. In my opinion, they don't address some of the problems within the community that need to be addressed:

- The tasks studied by many people in the reservoir computing community are not of interest to many outside of the nonlinear dynamics community. For example, the spoken digit task has little commercial interest.

To this end, they should suggest a mechanism for the community to identify problems that are of interest and then partner with industry to develop/curate datasets that can be processed with various algorithms.

- Along a similar line, most people do not care about dynamical systems that display chaotic behaviors. One possible counterexample is simulating full 3d turbulence, normally the domain of computational fluid dynamics. Can a reservoir computer be used to drastically reduce the computation for these expensive simulations? NAVASTO GmbH already has a product doing this, but it is not clear the neural network architecture underlying their algorithm.

Another example application that might make some impact is learning neuronal dynamics in the brain. The datasets being collected by neurobiologists are expanding rapidly and this will be an emerging opportunity.

Anyway, pointing out problems where the world <does> care about complex dynamics should be made.

- The authors mention application to control. But there is a mismatch between what the community has done and what will be acceptable to the control community and to government agencies that need to certify control algorithms where human safety is an issue.

First, control guarantees have to be established and this had not been done by researchers in the reservoir computing community. Second, the algorithms have to be explainable. For example, if the algorithm fails for some reason, the government agencies will require a method to trace back and explain why it failed.

- The other issue not addressed is why a practitioner using deep learning networks should consider reservoir computing. There have been a few works doing a direct comparison between deep learning networks with long short-term memory and reservoir computers. See several papers from Ed Ott's group at the University of Maryland. A short discussion of this comparison would be useful.

- As mentioned by the authors, a reservoir computer is a universal approximator of functions. So is a deep feedforward network. So why can't a reservoir computer be as successful as a deep learning network? This should be given as a challenge to the reservoir computing community. Specific tasks are natural language processing, image recognition, etc. These are applications that the commercial sector cares about. Can't the community figure out what has been holding reservoir computing back?

Some in the deep learning community claim that a reservoir computer is limited because of the randomly assigned weights. However, this point has been addressed in the current works on representation theory. This point should be stressed more strongly.

Addressing this question would have profound implications. The amount of electricity consumed training deep learning models is considerable. Finding a way to make a reservoir computing model work for these applications would greatly advance the field of machine learning and have huge societal impact.

Reviewer #2 (Remarks to the Author):

The manuscript 'Emerging Opportunities and Challenges for the Future of Reservoir Computing' by Yan et al. is the combination of an overview and roadmap/outlook article on the concept of reservoir computing (RC). It gives the reader a complete, sound general introduction to the basics of RC, the related underlying theoretical concept to interpret its performance and to understand its general potential as well as limitations. This is then cast in the light of implementation of RC in physical hardware

substrates in general, which is one of the major appeals of the concepts. Such hardware implementations are particularly displayed by the realizations of photonic reservoirs, which is a strong community, and the authors provide a general overview of the historic evolution in the field, highlighting several key moments where crucial sections have been demonstrated for the first time.

The field of RC and its implementations in hardware substrates is a highly relevant and dynamic scientific community. The generally agnostic / tolerant nature of RC against specificities of the physical substrates nature, i.e. electronic, photonic, biological, mechanics etc., makes RC highly relevant for the general physics and engineering community. As such the manuscript is an excellent fit to the audience of Nature Communications. The manuscript is excellently written, and I only have some punctual suggestions regarding its content plus a couple of comments regarding citations. In general, I can recommend the manuscript for publication after these changes have been implemented.

My two main comments are concerning (i) a potentially missed opportunity regarding timescales and the related opportunities / challenges of physical substrate RC, and (ii) a slightly more critical discussion of the training concepts and the related hardware implementations of readout weights.

(i) Timescales and the associated challenges / opportunities

The authors discuss a variety of potential hardware implementations, where they highlight the possibilities of efficient and ultra-fast applications based on physical substrate-based RC. I fully heartedly agree, yet there is an Elephant in the room. For this to happen efficiently and to significantly exploit the RC hardware concept, one needs to match the timescales between the computational challenge and the internal dynamics of the RC substrate. Otherwise either (a) the substrate response is far too small in the case of too fast timescales in the problem compared to the RC substrate, or (b) the fading memory of RC is not given. This is a quite fundamental aspect, and I think in particular with the various application examples provided by the author, the manuscript could substantially gain in relevance for the general audience if these opportunities are discussed also in light of their timescales, providing maybe also a link to suited physical processes to implement such a RC in hardware.

(ii) Training and readout weights

At various positions the ease of training a RC is highlighted. However, once a RC is to be implemented in hardware, most of these advantages have to be seen with a bit more scrutiny. In most settings, it is not realistic to sample and store a large variety of node-responses to a certain input due to either sampling bandwidth restrictions, memory depth and bandwidth, or all combined. It is simply not possible in many cases to probe a system with 10s-1000s of probes, interfaced with AD converters. Besides these practical challenges, almost always this will require auxiliary hardware of such complexity that the attractiveness of the RC hardware is abolished in the first place. Finally, the usual drift found in hardware requires under most realistic settings that this procedure has to be repeated quite regularly, hence it cannot be a one-of optimization. In fact, one of the few systems where readout weights implemented in hardware under

realistic settings have been demonstrated are delay RC computing [Antonik2016, DOI: 10.1109/TNNLS.2016.2598655], with the associated questionable benefit of delay systems as discussed also by the authors in the manuscript. As well as in a spatio-temporal system that actually implements the entire concept for the first time scalable and demonstrated realtime training [Porte2021, 10.1088/2515-7647/abf6bd].

Neither article is cited, and actually I think the topic and these articles should feature quite prominently given the importance and the remaining challenge of this crucial step for the success of the entire concept.

Some minor points:

line 46: 'promising. RC conceptualizes how a brain-like system operates, ...' I find this statement far too general and hence misleading.

line 53: 'including its input and processing layers, ...' this is a very narrow support for the previous general claim, and also a reference should be provided.

line 55: 'in some optimal way' -> 'in some optimized'?

line 92: dynamical systems are usually described in the form ' $dx = -x(t) + F(\dots)$ '

line 103: 'optimization', typo

Figure 2: the first demonstration of an electro-optical delay RC was done by Larger et al, not by reference [27]. They both were published in close succession, so if the authors insist then both references can be given, but currently this is not the correct citation.

Line 365: 'limit; For electrical' I am not sure if that semi-colon and capital letter afterwards is correct.

Table 1: Currently the free-space reference gives kHz as the speed limit, which is not correct considering the publication by Porte et al., see point (ii) of my main comments, that has the straight forward potential to GHz realtime inference bandwidths.

Line 397: 'on. The super power of traditional neural networks' I find this formulation using 'super power' a bit alienating

Line 475: 'WER is reaching near-perfect levels (0.4%) [33].' Brunner et al (2013) in [56] demonstrated a WER of 0.014%.

Reviewer #3 (Remarks to the Author):

The paper is a review on reservoir computing, a promising computing framework, particularly well suited for physical implementation. The paper is well written, and bring the reader from the fundamental mathematical framework, to the physical implementations, the various tasks it can solve, and propose a forward looking perspective about its future possible applications.

The paper is well written, well illustrated, and constitute a nice introduction to the field. It appears to me up to date, and scientifically sound. I am therefore supportive of publication.

One general recommendation is maybe to strengthen a bit the connection between reservoir computing and the general field of neuromorphic computing, which is only alluded to.

Other than that, I just have a few minor remarks and suggestions :

-L137: « echo-state » property is only defined later, and is mysterious at this stage of the manuscript.

-L220 : « for » should be removed for grammatical correctness

-L262 : « Due to the simulation characteristics of the system « is a bit confusing, maybe reformulate and/or connect to neuromorphic computing?

L362: « due to its instruction free and shared memory free architecture « maybe mention the non-von neumann / in-memory nature of the computing. ^[1]_[SEP]

L370 : « And it also faces... » it is unusual to start a sentence by « and » so the articulation with the rest is not really clear.

L397 : « The super power of traditional neural networks in classification tasks has been verified in lots of work. « is a bit colloquial / general, maybe rephrase?

Fig 4: there is only one reference mentioned in the caption (ref 19), I would make sure all works displayed on the figure are properly referenced, either in the figure directly or in the caption.

Here and there : « it's » to replace by « it is » and some other colloquial expressions to maybe amend.

Summary of Main Changes to the Manuscript

Figures and BOX.

1. BOX1 is added to illustrate the differences and connections between deep learning neural network and reservoir neural network.
2. Fig. 1 is now incorporated as part of BOX2; Fig. 3 is now incorporated as part of BOX3.
3. We improved original Fig. 4 and Fig. 5 (appear as new Fig. 2 and Fig. 3, respectively.)

Main Text

1. Section 5 has been reorganized to clearly show “potentials” and “challenges” as separate parts. In addition, original subsection “Industrial simulation and digital twins” is renamed to “AI for science and digital twins”. A new paragraph is added at last in this subsection, to discuss about other complex dynamical systems in the real world.
2. ‘Discussion’ is now a standalone part. In the revised Discussion section: (1) A new paragraph is introduced (now at the end of the section) to discuss potential mechanisms for the research community to identify problems that are of interest and then partner with industry to develop and curate datasets that can be processed with various algorithms. (2) The original first and second paragraph are combined to the new first paragraph, to discuss challenges in theoretical and algorithmic levels of RC. (3) The original third paragraph now is second paragraphs, we rewrite it to illustrate the physical timescales, training and readout challenges associated with RC.

References added - Based on the referees suggestions, we have included a few additional references into the revised manuscript, as follows (indexed as in the revised manuscript):

- [9] Schuman, C. D., Kulkarni, S. R., Parsa, M., Mitchell, J. P., Date, P., & Kay, B. (2022). Opportunities for neuromorphic computing algorithms and applications. *Nature Computational Science*, 2(1), 10-19.
- [10] Christensen, D. V., Dittmann, R., Linares-Barranco, B., Sebastian, A., Le Gallo, M., Redaelli, A., ... & Pryds, N. (2022). 2022 roadmap on neuromorphic computing and engineering. *Neuromorphic Computing and Engineering*, 2(2), 022501.
- [67] Larger, L., Soriano, M. C., Brunner, D., Appeltant, L., Gutiérrez, J. M., Pesquera, L., ... & Fischer, I. (2012). Photonic information processing beyond Turing: an optoelectronic implementation of reservoir computing. *Optics express*, 20(3), 3241-3249.
- [87] Porte, X., Skalli, A., Haghighi, N., Reitzenstein, S., Lott, J. A., & Brunner, D. (2021). A complete, parallel and autonomous photonic neural network in a semiconductor multimode laser. *Journal of Physics: Photonics*, 3(2), 024017.
- [139] Wang, K., Shen, Z., Huang, C., Wu, C. H., Eide, D., Dong, Y., ... & Rogahn, R. (2019). A review of microsoft academic services for science of science studies. *Frontiers in Big Data*, 2, 45.
- [140] Smolensky, P., McCoy, R., Fernandez, R., Goldrick, M., & Gao, J. (2022). Neurocompositional computing: From the Central Paradox of Cognition to a new generation of AI systems. *AI Magazine*, 43(3), 308-322.
- [141] Callaway, E. (2020). 'It will change everything': DeepMind's AI makes gigantic leap in solving protein structures. *Nature*, 588(7837), 203-205.
- [142] Callaway, E. (2022). The entire protein universe': AI predicts shape of nearly every known protein. *Nature*, 608(7921), 15-16.

- [143] Lee, P., Bubeck, S., & Petro, J. (2023). Benefits, limits, and risks of GPT-4 as an AI chatbot for medicine. *New England Journal of Medicine*, 388(13), 1233-1239.
- [144] Hu, Z., Jagtap, A. D., Karniadakis, G. E., & Kawaguchi, K. (2023). Augmented Physics-Informed Neural Networks (APINNs): A gating network-based soft domain decomposition methodology. *Engineering Applications of Artificial Intelligence*, 126, 107183.
- [145] Kashinath, K., Mustafa, M., Albert, A., Wu, J. L., Jiang, C., Esmailzadeh, S., ... & Prabhat, N. (2021). Physics-informed machine learning: case studies for weather and climate modelling. *Philosophical Transactions of the Royal Society A*, 379(2194), 20200093.
- [157] Amil P, Soriano M C, Masoller C. Machine learning algorithms for predicting the amplitude of chaotic laser pulses[J]. *Chaos: An Interdisciplinary Journal of Nonlinear Science*, 2019, 29(11).
- [158] Antonik, P., Dupont, F., Hermans, M., Smerieri, A., Haelterman, M., & Massar, S. (2016). Online training of an opto-electronic reservoir computer applied to real-time channel equalization. *IEEE Transactions on Neural Networks and Learning Systems*, 28(11), 2686-2698.
- [159] Amir Gholami, Zhewei Yao, Sehoon Kim, Michael W Mahoney, and Kurt Keutzer. Ai and memory wall. RiseLab Medium Post, 2021.
- [160] Dai, Y., Yamamoto, H., Sakuraba, M., & Sato, S. (2021). Computational efficiency of a modular reservoir network for image recognition. *Frontiers in Computational Neuroscience*, 15, 594337.

*Changes in the main text are marked in **BLUE**.

Point-by-point Response to Referee 1's Comments

Referee 1: *The authors have prepared a concise summary of reservoir computing, tracing its history, success stories, and pointing out outstanding problems and likely future high-impact applications. This will likely be a nice article for researchers in the reservoir computing field.*

But it is not clear to me that the article will broaden the appeal of reservoir computers outside of the immediate community. In my opinion, they don't address some of the problems within the community that need to be addressed:

Response: We thank the reviewers for the positive evaluation of our work and the constructive comments. In the revised manuscript we have addressed all the issues raised by the referee, with a point-by-point response to the specific comments as follows.

Referee 1: *- The tasks studied by many people in the reservoir computing community are not of interest to many outside of the nonlinear dynamics community. For example, the spoken digit task has little commercial interest.*

To this end, they should suggest a mechanism for the community to identify problems that are of interest and then partner with industry to develop/curate datasets that can be processed with various algorithms.

Response: We appreciate this insightful and valuable suggestion, and fully agree that a mechanism for the community to identify problems that interest is much needed. To emphasize this, in the last two paragraphs of the Discussion Section, we add contents to illustrate these. In particular, we mention that we are planning to establish a more accessible platform for all researchers who are interested in reservoir computing, and we also suggest building an open-source community gathering researchers and scientists from different fields and backgrounds.

Changes made to the manuscript: In Section 6, we rewrote the last paragraph as:

“Looking ahead, unlocking the full potential of RC and neuromorphic computing in general is critical yet challenging. In fact, this goes beyond just putting out open-source codes or solve a few specific problems. Innovative ideas and interdisciplinary research formats are much needed. As concrete suggestions, researchers of the applied mathematics and nonlinear dynamics communities who have been the main players in RC will need to get close(r) to the mainstream AI applications and try to develop next-generation RC systems to compete in these scenarios where the value of application has been established and recognized by the industry. A good starting point can be open-source tasks and datasets such as Kaggle, and more generally to directly partner with industrial research labs to put RC into real applications. On the other hand, raising awareness of the (potential) utility of RC requires attracting interest from researchers and decision-makers who are traditionally outside of the field. For instance, themed conferences and workshops may be organized to foster such discussions among scientists and researchers from diverse fields across academia and industry. Despite the many challenges, with persistence and innovations a new and future paradigm of intelligent learning and computing may possibly emerge from the works of RC and neuromorphic computing.”

Referee 1: - *Along a similar line, most people do not care about dynamical systems that display chaotic behaviors. One possible counterexample is simulating full 3d turbulence, normally the domain of computational fluid dynamics. Can a reservoir computer be used to drastically reduce the computation for these expensive simulations? NAVASTO GmbH already has a product doing this, but it is not clear the neural network architecture underlying their algorithm.*

Another example application that might make some impact is learning neuronal dynamics in the brain. The datasets being collected by neurobiologists are expanding rapidly and this will be an emerging opportunity.

Anyway, pointing out problems where the world <does> care about complex dynamics should be made.

Response: We appreciate these valuable suggestions, and we fully agree that pointing out problems where the world <does> care about complex dynamics should be made. For example, simulating full three-dimensional turbulence, weather forecasting, laser dynamics, and so on. To enhance this perspective, in the modified manuscript, in Section 5, we rename the subsection “Industrial simulation and digital twins” to “AI for science and digital twins”, and add specific illustrations.

Changes made to the manuscript: In Section 5, we added illustrations at paragraph “AI for science and digital twins”. In “*Opportunities*” part: “For example, simulating full three-dimensional turbulence, weather forecasting, laser dynamics, etc. Preliminary studies suggest that reservoir computing can be used to reduce the computational resources required for these expensive simulations. Arcomano, Troy, et al. \cite{arcomano2020machine} developed a low-resolution global prediction model based on reservoir computing and investigated the applicability of RC in weather forecasting. They demonstrated that a parallel ML model based on RC can predict the global atmospheric state in the same grid format as the numerical (physics-based) global weather forecast model. They also found that the current version of the ML model has potential in short-term weather forecasting. They further discovered that when full-state dynamics are available for training, RC outperforms the time-based backpropagation through time (BPTT) method in terms of prediction performance and capturing long-term statistical data while requiring less training time.

In “*Challenges*” part: “However, for reduced-order data, large-scale RC models may be unstable and more likely to exhibit bias than the BPTT algorithm. In another example of research on nonlinear laser dynamics, the authors found that RC methods have simpler training mechanisms and can reduce training time compared to deep neural networks [157]. For practical problems involving complex nonlinear physical processes, we have reason to believe that RC methods may provide us with solutions for computational acceleration.”

Referee 1: - *The authors mention application to control. But there is a mismatch between what the community has done and what will be acceptable to the control community and to government agencies that need to certify control algorithms where human safety is an issue.*

First, control guarantees have to be established and this had not been done by researchers in the reservoir computing community.

Second, the algorithms have to be explainable. For example, if the algorithm fails for some reason, the government agencies will require a method to trace back and explain why it failed.

Response: We completely agree. For control applications, it is in fact the case that a pure black-box controller is unlikely to be practically adopted for those very reasons pointed out by the referee. To clarify this important issue, we added sentences both in Section 4 (in the third last paragraph) when discussing control benchmarks and in Section 6 (in the first paragraph) when summarizing theoretical challenges of RC.

Changes made to the manuscript:

1. In Section 5, third last paragraph, we added the following sentence at end of the paragraph, “In MPC, since the role of RC merely replaces a linear predictor the overall controller architecture remains transparent and intact. In principle, it is possible to adopt RC for general controller design beyond usage in the MPC framework, e.g., directly learning control rules from data together with (some) prior model knowledge. However, the main challenge would be to pose theoretical guarantees on error and convergence neither of which have been resolved by existing works of RC.”
2. In Section 6, we added the following sentence in the first paragraph, “An important reason to further advance the mathematical theory of RC is for data-driven control applications. In most of those applications, rigorous theory on control error and convergence are necessary for the corresponding controller to be considered usable in an industrial setting. However, so far very little work has been done to address these important problems.”

Referee 1: - *The other issue not addressed is why a practitioner using deep learning networks should consider reservoir computing. There have been a few works doing a direct comparison between deep learning networks with long short-term memory and reservoir computers. See several papers from Ed Ott's group at the University of Maryland. A short discussion of this comparison would be useful.*

Response: Very good point. To more directly compare and contrast RC with deep learning, we made the following changes to the manuscript: (1) We created BOX to summarize the main differences and connections between deep learning and reservoir computing. (2) In the text paragraph within BOX1, we included a few new references: Ref. [159] lists specific size of parameters and required training flops of different networks in deep learning; Ref. [160] shows the training flops of reservoir computing in image recognition. These information give a relatively straightforward comparisons between deep learning and reservoir computing. Namely, reservoir computing has potentials to play roles when the training dataset is limited. In addition, we cite references [116, 117] from Ed Ott’s group. Reservoir computing has good performance in learning temporal data and make predictions. By inserting reservoir computing, Ed Ott’s work showed that the climate model can make more accurate predictions [116]. Besides, reservoir computing can achieve weather forecast task with shorter training time, compared with that of deep learning [117].

Referee 1: - *As mentioned by the authors, a reservoir computer is a universal approximator of functions. So is a deep feedforward network. So why can't a reservoir computer be as successful as a deep learning network? This should be given as a challenge to the reservoir computing community. Specific tasks are natural language processing, image recognition, etc. These are applications that the commercial sector cares about. Can't the community figure out what has been holding reservoir computing back?*

Response: Very good point and this is indeed one of the key motivating factors for this paper, to raise awareness of the potential of RC as well as pointing out the gaps. To further strengthen this perspective, we added a sentence to the very end of Section 4 after summarizing the current state of applying RC on real-world datasets.

Changes made to the manuscript: In the end of Section 4, the following sentence is added, “Despite attempts and preliminary successes in applying RC to problems endowed with real-world datasets, nearly none of those attempts have led to an industry-level adoption and application. An important reason is that the performance of RC on common tasks such as image classification, audio signal processing have not reached or shown to have the potential to approach the SOTA metrics offered by deep-learning based methods. Given that theoretically RC has universal approximation capacities just as general neural networks, in principle nothing seems to be holding back RC models to push the frontiers of most challenging AI tasks, and this should be a main goal of the entire RC community.”

Referee 1: *Some in the deep learning community claim that a reservoir computer is limited because of the randomly assigned weights. However, this point has been addressed in the current works on representation theory. This point should be stressed more strongly.*

Response: Good point. To better elucidate the state-of-the-art of the representation theory of RC and in particular how random weights themselves are justified, we expanded the following paragraph starting “Mathematical theory behind RC” in Section 2.

Changes made to the manuscript: The paragraph starting “Mathematical theory behind RC” in Section 2 is revised with the following sentence added to the end, “... In particular, despite having randomly assigned weights that are not trained, RC models are nevertheless shown to possess strong representation power with rigorous theoretical guarantees.”

Referee 1: *Addressing this question would have profound implications. The amount of electricity consumed training deep learning models is considerable. Finding a way to make a reservoir computing model work for these applications would greatly advance the field of machine learning and have huge societal impact.*

Response: We thanks referee’s constructive suggestions. We add discussions about connections and comparisons between deep learning and reservoir computing in the BOX1 to illustrate that reservoir computing has the potential to work comparably to deep learning models with smaller parameter size and training dataset in some fields, e.g., weather forecast and image recognition.

Point-by-point Response to Referee 2's Comments

Referee 2: *The manuscript 'Emerging Opportunities and Challenges for the Future of Reservoir Computing' by Yan et al. is the combination of an overview and roadmap/outlook article on the concept of reservoir computing (RC). It gives the reader a complete, sound general introduction to the basics of RC, the related underlying theoretical concept to interpret its performance and to understand its general potential as well as limitations. This is then cast in the light of implementation of RC in physical hardware substrates in general, which is one of the major appeals of the concepts. Such hardware implementations are particularly displayed by the realizations of photonic reservoirs, which is a strong community, and the authors provide a general overview of the historic evolution in the field, highlighting several key moments where crucial sections have been demonstrated for the first time.*

The field of RC and its implementations in hardware substrates is a highly relevant and dynamic scientific community. The generally agnostic / tolerant nature of RC against specificities of the physical substrates nature, i.e. electronic, photonic, biological, mechanics etc., makes RC highly relevant for the general physics and engineering community. As such the manuscript is an excellent fit to the audience of Nature Communications. The manuscript is excellently written, and I only have some punctual suggestions regarding its content plus a couple of comments regarding citations. In general, I can recommend the manuscript for publication after these changes have been implemented.

Response: We appreciate the reviewer's thorough and insightful comments, and we are glad that the manuscript is considered valuable and a good fit for the journal. In the revision we took the referee's suggestion and addressed all the issues raised to improve the manuscript.

Referee 2: *My two main comments are concerning (i) a potentially missed opportunity regarding timescales and the related opportunities/challenges of physical substrate RC, and (ii) a slightly more critical discussion of the training concepts and the related hardware implementations of readout weights.*

Response: We thank reviewer for the recognition and valuable suggestions. We totally agree that we should add the challenges/opportunities of the timescale between the computational challenge and the internal dynamics of the RC substrate. We also appreciate the reviewer for reminding the challenges of implementing training and readout weights in practical cases. We modified the related paragraph accordingly. In Section 6, in second paragraph, we discuss about matching the timescales between the computational challenge and the internal dynamics of the RC substrate. We also list some possible solutions in terms of adjusting the physical parameters of the RC substrate, adopting more complex RC architecture and with the help of other technologies. As for Training and readout weights, we also add illustrations to discuss about them. We agree that the two references proposed by referee in which implementing readouts physically instead of the software-based readouts is a possible solution, and the references are also cited.

Referee 2: (i) *Timescales and the associated challenges / opportunities*

The authors discuss a variety of potential hardware implementations, where they highlight the possibilities of efficient and ultra-fast applications based on physical substrate-based RC. I fully heartedly agree, yet there is an Elephant in the room. For this to happen efficiently and to significantly exploit the RC hardware concept, one needs to match the timescales between the computational challenge and the internal dynamics of the RC substrate. Otherwise either (a) the substrates response is far too small in the case of too fast timescales in the problem compared to the RC substrate, or (b) the fading memory of RC is not given. This is a quite fundamental aspect, and I think in particular with the various application examples provided by the author, the manuscript could substantially gain in relevance for the general audience if these opportunities are discussed also in light of their timescales, providing maybe also a link to suited physical processes to implement such a RC in hardware.

Response: We appreciate the reviewer for this valuable suggestion, and we fully agree that the timescale and associated challenges should be emphasized. In Section 6, we add contents to discuss about matching the timescales between the computational challenge and the internal dynamics of the physical RC substrate.

Changes made to the manuscript: In Section 6, We combine the original first and second paragraph to the new first paragraph, and rewriting the experimental challenges of RC in the new second paragraph:

At the experimental level, there are still some gaps when mapping the RC model to the physical experiment. The first is timescale problem of physical substrate RC: Matching the timescales between the computational challenge and the internal dynamics of the physical RC substrate is a key issue in reservoir computing. If the timescale of the problem is much faster than the response time of the physical system, the response of the reservoir will be too small or the fading memory of the reservoir will not be properly utilized, rendering the physical reservoir computing system ineffective. One intuitive solution is to adjust the physical parameters of the reservoir to match the timescale of the computational problem. This poses high requirements for the design of RC network structures and training algorithms. Using other technologies such as super-resolution and compressive sensing to overcome the resolution problem of single-point measurement and processing in RC systems may be a viable solution.

Referee 2: (ii) *Training and readout weights*

At various positions the ease of training a RC is highlighted. However, once a RC is to be implemented in hardware, most of these advantages have to be seen with a bit more scrutiny. In most settings, it is not realistic to sample and store a large variety of node responses to a certain input due to either sampling bandwidth restrictions, memory depth and bandwidth, or all combined. It is simply not possible in many cases to probe a system with 10s-1000s of probes, interfaced with AD converters. Besides these practical challenges, almost always this will require auxiliary hardware of such complexity that the attractivity of the RC hardware is abolished in the first place. Finally, the usual drift found in hardware requires under most realistic settings that this procedure has to be repeated quite regularly, hence it cannot be a

one-of optimization. In fact, one of the few systems where readout weights implemented in hardware under realistic settings have been demonstrated are delay RC computing [Antonik2016, DOI: 10.1109/TNNLS.2016.2598655], with the associated questionable benefit of delay systems as discussed also by the authors in the manuscript. As well as in a spatio-temporal system that actually implements the entire concept for the first time scalable and demonstrated realtime training [Porte2021, 10.1088/2515-7647/abf6bd]. Neither article is cited, and actually I think the topic and these articles should feature quire prominently given the importance and the remaining challenge of this crucial step for the success of the entire concept.

Response: We appreciate the referee for this valuable suggestion, the articles mentioned should indeed be by cited, and we fully agree that the training and readout challenges should be emphasized. We add the related discussions in Section 6, paragraph four and five to illustrate them, and the proposed references [87, 158] are also cited.

Changes made to the manuscript: In Section 6, in the new second paragraph, we add contents: The second is the real-time data processing problem: One of the significant advantages of reservoir computing is lightweight and fast computation. However, in practical physical systems, it is often unrealistic to sample and store a large number of node responses to a certain input due to limitations such as sampling bandwidth, storage depth and bandwidth, or their combinations. It is simply not feasible in many cases to probe a system with a large number of probes (10s-1000s) interfaced with AD converters. In addition to these practical challenges, hardware drift often requires regular repetition of calibration procedures, hence it cannot be a one-of optimization. Furthermore, data preprocessing and postprocessing also limit the overall computational speed of the physical RC system. One approach to address this issue is to use hardware-based readout instead of software-based readout [87, 158].

Referee 2: *Some minor points:*

line 46: 'promising. RC conceptualizes how a brain-like system operates, ...' I find this statement far to general and hence misleading.

Response: We add illustrations before this sentence to elucidate the relevance of RC in the more general field of neuromorphic computing. Please refer to first three sentences in the second paragraph of Introduction Section.

Referee 2: *line 53: 'including its input and processing layers, ...' this is a very narrow support for the previous general claim, and also a reference should be provided.*

Response: Thanks for the good suggestion. In the beginning of this paragraph, we add illustrations and references to generally explain the neuromorphic computing, and now the claim is more complete.

Referee 2: *line 55: 'in some optimal way' -> 'in some optimized'?*

Response: Fixed.

Referee 2: *line 92: dynamical systems are usually described in the form ' $dx = -x(t) + F(\dots)$ '*

Response: Here we use Δ to denote the change of x , and we explain it is dx in the continuous-time system, and $x(t + 1) - x(t)$ in discrete-time system in lines 92~93. So here we temporarily keep the Δ notation.

Referee 2: *line 103: 'optimization', typo*

Response: Fixed.

Referee 2: *Figure 2: the first demonstration of an electro-optical delay RC was done by Larger et al, not by reference [27]. They both were published in close succession, so if the authors insist then both references can be given, but currently this is not the correct citation.*

Response: We thank the reviewers for providing constructive comments. We modified the figure and corrected the citation to reference [67].

Referee 2: *Line 365: 'limit; For electrical' I am not sure if that semi-colon and capital letter afterwards is correct.*

Response: Fixed.

Referee 2: *Table 1: Currently the free-space reference gives kHz as the speed limit, which is not correct considering the publication by Porte et al., see point (ii) of my main comments, that has the straight forward potential to GHz realtime inference bandwidths.*

Response: Fixed

Referee 2: *Line 397: 'on. The super power of traditional neural networks' I find this formulation using 'super power' a bit alienating.*

Response: 'super power' is changed to 'effectiveness'.

Referee 2: *Line 475: 'WER is reaching near-perfect levels (0.4%) [33].' Brunner et al (2013) in [56] demonstrated a WER of 0.014%.*

Response: Fixed.

Point-by-point Response to Referee 3's Comments

Referee 3: *The paper is a review on reservoir computing, a promising computing framework, particularly well suited for physical implementation. The paper is well written, and bring the reader from the fundamental mathematical framework, to the physical implementations, the various tasks it can solve, and propose a forward looking perspective about its future possible applications.*

The paper is well written, well illustrated, and constitute a nice introduction to the field. It appears to me up to date, and scientifically sound. I am therefore supportive of publication.

Response: We are grateful for the reviewer's careful reading and encouraging comments.

Referee 3: *One general recommendation is maybe to strengthen a bit the connection between reservoir computing and the general field of neuromorphic computing, which is only alluded to.*

Response: We thank referee for this valuable suggestion. To strengthen the connection between RC and neuromorphic computing, we expanded the beginning of the second paragraph in Introduction, which now reads "As an alternative direction to the current deep learning paradigm...".

Changes made to the manuscript: In the Introduction section, the first sentence of the second paragraph is now revised and expanded as "As an alternative direction to the current deep learning paradigm, research into the so-called neuromorphic computing has been attracting significant interest [9]. Neuromorphic computing generally focuses on developing novel types of computing systems that operate at a fraction of the energy comparing against current transistor-based computers, often deviating from the von-Neumann architecture and drawing inspirations from biological and physical principles [10]. Within the broader field of neuromorphic computing, an important family of models known as RC has progressed significantly over the past two decades [11,12]."

Referee 3: *Other than that, I just have a few minor remarks and suggestions :*

Response: We thank the review for the careful reading the detailed suggestions, all of which have been addressed in this revision, as follows.

Referee 3: *-L137: « echo-state » property is only defined later, and is mysterious at this stage of the manuscript.*

Response: We thank referee's valuable suggestion. In order to maintain clarity of the content and concept, we chose to remove 'on the system having the "echo-state" property', leaving the description of "echo state property" in detail in the later part.

Referee 3: -L220 : « for » should be removed for grammatical correctness

Response: Fixed.

Referee 3: -L262 : « Due to the simulation characteristics of the system « is a bit confusing, maybe reformulate and/or connect to neuromorphic computing?

Response: We reformulate the sentence “Due to the simulation characteristics of the system,” to “As the development of integration technology for electrical and optical component,”.

Referee 3: L362: « due to its instruction free and shared memory free architecture « maybe mention the non-von neumann / in-memory nature of the computing.

Response: We reformulate the sentence “due to its instruction free and shared memory free architecture.” to “due to its non-Von Neumann/in-memory nature of the computing.”

Referee 3: L370 : « And it also faces... » it is unusual to start a sentence by « and » so the articulation with the rest is not really clear.

Response: Fixed.

Referee 3: L397 : « The super power of traditional neural networks in classification tasks has been verified in lots of work. « is a bit colloquial / general, maybe rephrase?

Response: Good suggestion. ‘super power’ is changed to ‘effectiveness’.

Referee 3: Fig 4: there is only one reference mentioned in the caption (ref 19), I would make sure all works displayed on the figure are properly referenced, either in the figure directly or in the caption.

Response: All cited references of the relevant figures are added in the caption.

Referee 3: Here and there : « it's » to replace by « it is » and some other colloquial expressions to maybe amend.

Response: Fixed.

REVIEWERS' COMMENTS:

Reviewer #1 (Remarks to the Author):

I think the authors have done an excellent job addressing the concerns of all referees. I support publishing the current version of the manuscript in Nat Comm.

Reviewer #2 (Remarks to the Author):

I thank the authors for their detailed and thorough replies to my comments. I can recommend the manuscript for publication.

Reviewer #3 (Remarks to the Author):

I am happy with the answers of the authors to my comment and I support publication.

Response to Referees Comments

REVIEWERS' COMMENTS:

Reviewer #1 (Remarks to the Author):

I think the authors have done an excellent job addressing the concerns of all referees. I support publishing the current version of the manuscript in Nat Comm.

Reviewer #2 (Remarks to the Author):

I thank the authors for their detailed and thorough replies to my comments. I can recommend the manuscript for publication.

Reviewer #3 (Remarks to the Author):

I am happy with the answers of the authors to my comment and I support publication.

Responses: We again thank the reviewers for their thorough evaluation in the previous round of the review which helped enormously for us to improve the manuscript. We believe this manuscript may inspire more scientists and researchers to notice RC and willing to have a exploration in related field.